# Experience of and access to maternity care in the UK by immigrant women: a narrative synthesis systematic review

Gina Marie Awoko Higginbottom ![ORCID],[1] Catrin Evans ![ORCID],[1] Myfanwy Morgan,[2] Kuldip Kaur Bharj,[3] Jeanette Eldridge,[4] Basharat Hussain[1]

[1]School of Health Sciences, University of Nottingham, Nottingham, UK
[2]Institute of Pharmaceutical Sciences, King's College London, London, UK
[3]Dept of Midwifery, University of Leeds, Leeds, UK
[4]Research and Learning Services, School of Health Sciences, University of Nottingham, Nottingham, UK

**Correspondence to**
Professor Gina Marie Awoko Higginbottom;
gawoko@hotmail.com

## ABSTRACT

One in four births in the UK is to foreign-born women. In 2016, the figure was 28.2%, the highest figure on record, with maternal and perinatal mortality also disproportionately higher for some immigrant women. Our objective was to examine issues of access and experience of maternity care by immigrant women based on a systematic review and narrative synthesis of empirical research.

**Review methods** A research librarian designed the search strategies (retrieving literature published from 1990 to end June 2017). We retrieved 45 954 citations and used a screening tool to identify relevance. We searched for grey literature reported in databases/websites. We contacted stakeholders with expertise to identify additional research.

**Results** We identified 40 studies for inclusion: 22 qualitative, 8 quantitative and 10 mixed methods. Immigrant women, particularly asylum-seekers, often booked and accessed antenatal care later than the recommended first 10 weeks. Primary factors included limited English language proficiency, lack of awareness of availability of the services, lack of understanding of the purpose of antenatal appointments, immigration status and income barriers. Maternity care experiences were both positive and negative. Women with positive perceptions described healthcare professionals as caring, confidential and openly communicative in meeting their medical, emotional, psychological and social needs. Those with negative views perceived health professionals as rude, discriminatory and insensitive to their cultural and social needs. These women therefore avoided continuously utilising maternity care.

We found few interventions focused on improving maternity care, and the effectiveness of existing interventions have not been scientifically evaluated.

**Conclusions** The experiences of immigrant women in accessing and using maternity care services were both positive and negative. Further education and training of health professionals in meeting the challenges of a super-diverse population may enhance quality of care, and the perceptions and experiences of maternity care by immigrant women.

## Strengths and limitations of this study

► Immigration is an international phenomenon, and this review increases understanding of how immigrant women navigate maternity services in the UK.
► The review systematically maps the positive and negative aspects of maternity care provision as experienced by immigrant women.
► The review provides strategic direction for enhancement of maternity care services.
► The review does not address the experiences of maternity care for second-generation women (eg, women of black and minority origin born in the UK).

## INTRODUCTION

The UK is in a period of superdiversity that is characterised by 'an increased number of new, small and scattered, multiple-origin, transnationally connected, socio-economically differentiated and legally stratified immigrants'.(Vertovec, p1024)[1] This presents challenges for the delivery and configuration of maternity services in achieving equality of provision which forms a key aim of the National Health Service (NHS) in the UK.[2] One in four births in the UK is to foreign-born women.[3] Indeed some immigrant women (depending of country of origin) appear disproportionately in confidential inquiries into maternal and perinatal mortality,[4] perhaps indicating possible deficits in the delivery of care, access and utilisation. Our review contributes to amelioration of this situation by synthesising knowledge related to maternity care access and interventions so as to configure appropriate interventions as identified per the NHS Midwifery 2020 vision to guide professional development of healthcare professionals (HCPs).[5] Reshaping care to ensure culturally safe and congruent maternity care that will not only benefit immigrant women but also improve the health of future generations in the UK.[3 4 6] Without the delivery of culturally appropriate and culturally safe maternal care, negative event trajectories may occur that range from simple miscommunications to life-threatening incidents,[7–9] risking increased maternal and

perinatal mortality. While recent reviews have focused on specific aspects of maternity care,[10 11] they have not considered a comprehensive conceptualisation of access or the current super diversity and redesign of NHS maternal services to meet the needs of immigrant women which requires integration of all these aspects.[2] We have addressed this deficit in our current review which utilises Gulliford *et al*'s theory of access to care.[12]

Considering the global context, some commonality exists between high income nations in the maternity care experiences of immigrant women: studies in the USA,[13] Canada,[11] Australia,[14 15] Sweden[16 17] and Germany,[8 18] all provided evidence of this in earlier international reviews led by Higginbottom *et al*[7 19] and Gagnon *et al*.[11] However, the international comparative reviews by Gagnon focused on specific populations of South Asian and Somali women in the UK[11] which form established immigrant groups rather than the more recent super diverse patterns of migration. We have addressed this deficit in our current review. Wehave addressed this deficit in our current review.

## CONCEPTUAL DEFINITIONS

There is no consensus definition in the UK regarding the definition of the term immigrant[20] with the terms immigrant and migrant which are frequently used interchangeably across different data sources and datasets whilst conveying the same meaning. *Country of birth* is used by The Annual Population Survey of workers and Labour Force Survey as a precursor for defining a 'migrant'. This survey therefore declares a person born outside the UK is classified as a 'migrant'. Noteworthy is the fact that workers born outside the UK may become British citizens with increasing residence in the UK.

A second source of data on migrants is applications made to obtain a National Insurance Number. This differs from the former in that the term migrant is conferred on the basis of *nationality*. All applicants who hold nationality other than the UK are therefore considered migrants. However, the situation is dynamic in that the nationality of a person may also to change over time and in some cases individuals may acquire dual citizenship involving several nation states.

A third and significant source of data on migrants is the Office for National Statistics (ONS). ONS utilises a different strategy classification which focuses on the notion of short-term international migrant and long-term international migrant. In this definition, the term 'long-term' refers to holding the intention of residing longer than a year, whereas short-term is intention of residing less than a year. The implication of this is that the ONS considers *length of stay* of a person in the UK as critical in determining migrant status which reflects the United Nations (UN) recommended classification of migrant into short and long term. Additionally, ONS utilises the UN definition of long-term international migrant. Accordingly, 'a migrant is someone who changes his or her country of usual residence for a period of at least a year, so that the country of destination effectively becomes the country of usual residence'.[20] In long-term international migration data, students and asylum-seekers are also included which differs for example from the situation in the USA.

## Immigrants and the UK NHS

In respect of service provision, the NHS adheres to the mandates set by central government that determines immigrant's entitlement to free NHS care. These mandates are concerned with the immigrant status and the type of service provision.[21] Within these mandates, an asylum-seeker woman may not be entitled to full maternity care because of immigration status.[22] Moreover, data collection by the NHS on this topic is not well established or comprehensive. Currently, the NHS usually collects data on ethnicity and nationality and not on migration-related variables such as length of stay, country of origin and so on.

The National Institute for Health and Care Excellence (NICE) which provides clinical guidelines for healthcare practice in the UK see NICE (2010)[23] identified recent migrant women as having complex social needs in its guidelines on *Pregnancy and complex social factors: a model for service provision for pregnant women with complex social factors* identified recent migrant women having complex social needs. Within the NICE definition, a recent migrant woman is a woman has who moved to the UK within the previous 12 months. This generic definition of the term migrant conflates migrant women of all classifications (eg, economic migrants, asylum-seekers, refugees and those lacking English language proficiency). This suggests that there is implicit acceptance of the term migrant women in healthcare in respect of being born outside the UK, and being subject to immigration regulations, together with possible challenges in English language proficiency.

## The operational definition of an immigrant women used in this review

The preceding paragraphs suggests that the term 'immigrant' is defined in various ways in different countries and by different authors. However, two features are frequently referred to in these definitions, namely 'country of birth' and 'length of stay'. These factors are noted by the NICE guidelines[23] on the provision of maternity care as important in entitlement, access and ability to use healthcare in the UK. For example, if you are born outside the UK, it is unlikely that you are knowledgeable about the UK healthcare provision.

We adopted the following definition of an immigrant woman for the purposes of our review, and most importantly, to inform our inclusion and exclusion criteria. We defined a woman as an immigrant if she was:
► Born outside the UK.
► Living in the UK for more than 12 months or had the intention to live in the UK for 12 (or more) months when first entered.

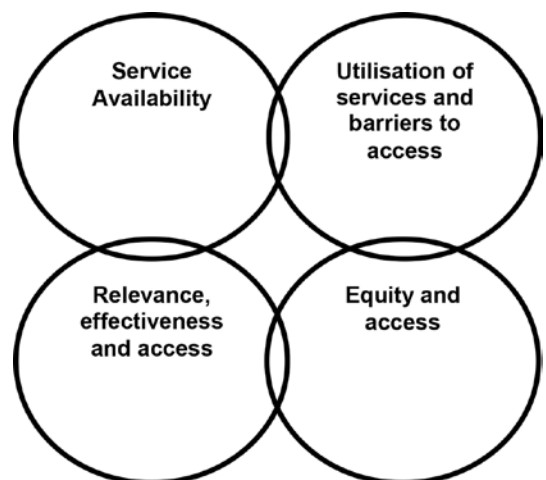

**Figure 1**   Gulliford *et al's* theory of access.

We therefore included studies on immigrant women where the population studied fulfils these two characteristics and included population groups of foreign students, asylum-seekers, recent legal refugees and immigrants, and illegal immigrants. In cases where the study populations/sample was not accurately or fully described, we employed the criteria of linguistic ability, as demonstrated by the need for an interpreter as a proxy for immigrant status. Notwithstanding all of these perspectives, we acknowledge that the term 'immigrant women' is generic and refers to a highly heterogeneous group of individuals with a complex and vast array of ethnocultural groups.

### Aim and rationale

We consider in this paper how accessibility and acceptability manifest, as important dimensions of access to maternity care services in terms of women's perception about availability of services and their experiences of accessing these services. We also consider whether evaluated interventions exist that challenge inequalities in maternity healthcare provision.

Our review employed two theoretical frameworks. These are Gulliford and colleagues' theory of access and second the concept of cultural safety.

A theory of access to services developed by Gulliford *et al*[12] map out four dimensions (figure 1):

1. Service availability.
2. Utilisation of services and barriers to access (which includes personal, financial and organisational barriers).
3. Relevance, effectiveness and access.
4. Equity and access.

We used this theoretical model in our systematic review which was based on a synthesis project funded by the National Institute for Health Research. Unlike most access models in the USA, this framework reflects the philosophy of the NHS in that its key principles are to provide horizontal access in terms of ensuring equality of access in the population and to achieve vertical access in terms of meeting the needs of particular groups in the population, such as minority ethnic groups. The application of these principles is influenced by availability,

accessibility and acceptability. The Gulliford model[12] has been widely used in empirical research, with the main paper cited over 730 times. This model with its emphasis on accessibility, acceptability, relevance and effectiveness, is entirely appropriate for assessing the provision of maternity services to minority ethnic groups and was employed in this review to assist in initial theme development and to examine how this access model intersected with our evidence.

Second, concepts of cultural safety provided a theoretical lens for the production of recommendations. Cultural safety is a theory that aims to assist the understanding of deficits in care by considering the historical and social processes that impact power relationships within and beyond healthcare.[24] Cultural safety is achieved when programmes, instruments, procedures, methods and actions are implemented in ways that do not harm any members of the culture or ethnocultural group who are the recipients of care. Those within the culture are best placed to know what is or is not safe for their culture which suggests the need for increased dialogue about immigrant and partner approaches.[25–29]

### METHODS

We employed Popay's approach to Narrative Synthesis (NS)[30] which consists of four elements (for a comprehensive explanation please see our published protocol.[31] The unique feature of this approach is that it provides highly specified steps.

Team members have successfully employed NS previously and have vast expertise in its usage.[7]

► Element 1: Developing a theory of why and for whom.
► Element 2: Developing a preliminary synthesis of the findings of the included studies, following implementation of the search strategy.
► Element 3: Exploring relationships in the data.
► Element 4: Assessing the robustness of the synthesis.

TheNS approach relies primarily on text to summarise the findings and produce asynthesis of the narrative findings of included papers. NS may be used with all paradigms of research quantitative, qualitative studies and mixed methods research studies, as the emphasis is on an interpretive synthesis of the narrative findings of research rather than on a metadata analysis.[30]

### Search strategy refinement and implementation

The search strategy employed key terms used in consistently formulated text-based queries and search statements. These terms were based on subject headings, thesaurus terms or related indexing and categorisation terms appropriate for each literature database. An example of a detailed final search strategy is given in online supplementary file 1. First, we searched 10 electronic databases using the aforementioned strategies (online supplementary file 2*)*. Following this, we searched for appropriate grey literature in SI Web of Knowledge Conference Proceedings Citation Index (Science 1990–),

Ovid MEDLINE 1948– and MEDLINE in-process and other non-indexed citations to daily update
► Ovid EMBASE 1980–2017 week 11
► Ovid PsycINFO 1972–March week 3 2017
► CINAHL Plus with full text/EBSCOHost to 2017
► MIDIRS on Ovid 1971 to April 2017
► Thomson Reuters Web of Science* 1900–2017
► ASSIA on ProQuest 1987–current
► HMIC on Ovid 1979–January 2017
► POPline (via http://www.popline.org/) 1970 to the present
Thomson Reuters Web of Science 1900–2017 includes the following:
► Science Citation Index Expanded 1900–2017
► Social Sciences Citation Index 1956–2017
► Conference Proceedings Citation Index-Science 1990–2017
► Conference Proceedings Citation Index-Social Science and Humanities 1990–2017
► Book Citation Index-Science 2008–2017
► Book Citation Index-Social Science and Humanities 2008–2017
► Emerging Sources Citation Index - 2015–2017

ISI Web of Knowledge Conference Proceedings Citation Index (Social Science and Humanities 1990–), ProQuest Dissertations and Theses, and the Cochrane Methodology Register. We also searched using Google and Google Scholar and consulted with the study expert advisory group. In conclusion, we hand searched the reference list of all included studies and relevant systematic reviews. Citations were downloaded into an ENDNOTE library, and following this all duplicates removed. The bibliographic databases that we searched are listed in box 1.

We adopted the PICO approach to implement the search strategy as follows:

P=immigrant women

I=maternity care

C=non-immigrant women—implicit comparator emerging in the results

O=experience of care

Our search strategy development was therefore based on:

Search concept 1=pregnancy, childbirth (implicitly females requiring maternity care), explicit terms covering women/females requiring all types of maternity care (antenatal, perinatal, postnatal, etc).

Search concept 2=immigrant populations (which would not fully distinguish between 'new' and 'second-generation' immigrants—this would be done at the selection stage).

Search concept 3=terms used to identify access to, use of, deficiencies in and so on, service provision (to help identify groups with poorer health outcomes or vulnerabilities)

This comprehensive search strategy generated high rates of retrieval of records, however many were not pertinent.

### Screening for relevance

In many cases, the study populations/sample was not fully described. In this situation, we contacted the authors for further clarification and in some cases used linguistic ability, for example, the need for an interpreter as a *proxy for immigrant status*. Our focus was on first-generation immigrant women regardless of their phenotype which led to inclusion of women of *white ethnicities*, although we encountered few studies that focusing on these groups. Study screening was undertaken independently by two team members (GH & BH) who employed our screening tool to assess the relevance of titles and abstracts in respect of our screening tool. The entire team reviewed papers classified as ambiguous papers in order to achieve a consensus agreement and where necessary full text papers of potentially included studies were retrieved and appraised. The exclusion and inclusion criteria can be found in online supplementary file 3. When we retrieved full-text papers which were later rejected, we have documented these excluded papers and presented a rationale for exclusion. These can be found in online supplementary file 4.

### RESULTS

#### Studies included in the review, findings and evidence

Our systematic review identified 40 empirical research studies in the scientific and grey literature. The included studies embraced a broad range of ethnocultural groups and methodological genres (see table 1 for master table of included studies and online supplementary file 5). The search outcomes are comprehensively detailed in figure 2, the Preferred Reporting Items for Systematic Reviews and Meta-Analyses flow chart.[32] The distribution of the studies across the themes are shown in figure 3 and publication dates in figure 4.

#### Data extraction and assessment of relevance

We conducted the following foundational activities in order to extract data (discussed in detail later).

*1) Textual description.* A systematic textual narrative was written for each study. We used headings adapted from Popay *et al*

Setting, Participants, Aim, Sampling and Recruitment, Method, Analysis, Results.[30]

*(2) Tabulation and summarisation of all studies to be included.* These tables described the attributes of the studies and the results. Information was extracted from the textual description using the same headings as above and additional headings as necessary. Papers in the PDF format were imported into ATLAS.ti qualitative data analysis software (ATLAS.ti Scientific Software Development, Berlin) using the 'Attributes' option to allow the tabulation of relevant data.

#### Quality assessment

In element 4, we conducted the quality appraisal (see tables 2 and 3).[33] All included studies were critically

**Table 1** Master table of included studies

| Reference | Study aim | Region | Methodology | Theory or framework | Setting | Data analysis | Sample and mode of recruitment |
|---|---|---|---|---|---|---|---|
| 2 | To provide insights into possible causes of poor maternity outcomes for new migrants in the West Midlands region of the UK and to develop recommendations that could help improve maternity services for these migrants. | West Midlands | Mixed methods: a semistructured questionnaire and in-depth interviews. | Not specified. | Not specified. | Qualitative: systematic thematic approach. Quantitative: triangulation of the findings. | A non-probability purposive sample was generated by selecting 82 women who had moved to the UK within the past 5 years and had subsequently utilised maternity services. Of these, 13 underwent in-depth interviews as well. |
| 17 | To apply the 'three delays' framework (developed for low-income African contexts) to a high-income Western scenario to identify delay-causing influences in the pathway to optimal facility treatment. | Greater London | Qualitative: individual and focus group interviews. | 'Three delays' framework. | Not specified. | Constructivist hermeneutic naturalistic study. | Purposive and snowball sampling was used to recruit 54 immigrant women originally from sub-Saharan regions in Africa (Somalia, Ghana, Nigeria, Senegal and Eritrea) living in London and to recruit 32 maternal providers. |
| 40 | To identify any social or ethnic differences in access to antenatal care and to quantify the effects of any such differences using data collected in a survey of women's experiences of antenatal screening. | England | Quantitative: a cross-sectional survey using a postal questionnaire. | Not specified. | Not specified. | Quantitative: cross-sectional analysis. | A stratified clustered random sampling strategy was used. Hospitals in England were stratified according to ethnic mix. To ensure inclusion of an adequate number of women from black and minority ethnicity (BME) backgrounds, hospitals with ≥15% of women of BME origin were oversampled. Pregnant women aged ≥16 years and receiving care in 15 participating hospitals were sent a postal questionnaire at 27–31 weeks of gestation. |
| 41 | To establish efficacy of link-worker services (an intervention) introduced for non-English-speaking Asian women in multiracial health districts. | Not specified | Quantitative survey: 21-item questionnaire. | Not specified. | | Qualitative: content analysis. | Questionnaire to the Heads of Midwifery Services in 30 multiracial district health authorities. 20 responded. Sample is not immigrant women, however this is an evaluation of an intervention. |
| 42 | To compare the health behaviours both antenatally (smoking and alcohol consumption) and postnatally (initiation and duration of breast feeding) of mothers who have white British or Irish heritage with those of mothers from ethnic minority groups and to examine in mothers from ethnic minority groups whether indicators of acculturation (generational status, language spoken at home and length of residency in the UK) were associated with these health behaviours. | England | Quantitative: a prospective nationally representative cohort study. | Not specified. | Not specified. | Quantitative: cohort study. | Stratified clustered sampling framework to over-represent mothers from ethnic minority groups and disadvantaged areas produced 6478 white British or Irish mothers and 2110 mothers from ethnic minority groups. Of those from ethnic minority groups, 681 (33%) were first generation and 55 (4%) second generation. |
| 43 | To determine the pregnancy outcomes of women of similar parity and ethnic background who received antenatal care ('booked') compared those who did not ('unbooked') over a period of 18 months. | North Middlesex University Hospital, London | Quantitative: a retrospective cohort study from September 2006 to March 2008 comparing the sociodemographics and the foetal and maternal outcomes of pregnancies of unbooked vs booked women. | Not specified. | Not specified. | Quantitative: a retrospective cohort study. | Women who received no antenatal care or who delivered within 3 days of their initial booking visit were categorised as 'unbooked'. In each case, the woman who had delivered next on the labour ward register (matched for ethnicity and parity) and who had received antenatal care prior to the second trimester served as a comparison. |
| 44 | To identify predictors of late initiation of antenatal care within an ethnically diverse cohort. | Newham, East London | Quantitative: a cross-sectional analysis of routinely collected electronic patient records from Newham University Hospital NHS Trust (NUHT). | Not specified. | Not specified. | Quantitative: cross-sectional analysis. | All women who attended their antenatal booking appointment within NUHT between 1 January 2008 and 24 January 2011 were included in this study. The main outcome measure was late antenatal booking, defined as attendance at the antenatal booking appointment after 12 weeks (+6 days) gestation. The sample included women from Somalia, Eastern Europe, Africa, the Caribbean and South Asia. |

Continued

**Table 1** Continued

| Reference | Study aim | Region | Methodology | Theory or framework | Setting | Data analysis | Sample and mode of recruitment |
|---|---|---|---|---|---|---|---|
| 45 | To compare the maternal and birth outcomes of Polish and Scottish women having babies in Scotland and to describe any differences in clinical profiles and service use associated with migration from Poland. | All over Scotland | Quantitative: a population-based epidemiological study of linked maternal country of birth, maternity and birth outcomes. Scottish maternity and neonatal records linked to birth registrations were analysed for differences in modes of delivery and pregnancy outcomes between Polish migrants and Scots. These outcomes were also compared with Polish Health Fund and survey data. | Not specified. | Not specified. | Quantitative: statistical analysis. | The study analysed 119 698 Scottish and 3105 Polish births to primiparous women in Scotland in 2004–2009 using routinely collected administrative data on maternal country of birth and birth outcome. |
| 46 | To determine the nature of the barriers confronting women when they used antenatal and postnatal services. | Pollokshields, Glasgow | Qualitative: semistructured questionnaire. | Not specified. | Not specified. | Qualitative: thematic analysis. | Twenty women were interviewed in depth by a Centre's Health Development Worker. Of these, 17 were born outside the UK. |
| 47 | To determine the current clinical practice of maternity care in England, including the service provision and organisations that underpin care, from the perspective of women needing the care; to identify the key areas of concern for women receiving maternity care in England; and to determine whether and in what ways women's experiences and perceptions of care have changed over the last 10 years. | England: not specified | Quantitative: survey. | Not specified. | Survey: not specified. | Quantitative: cross-sectional design. | Random samples of women selected for the pilot and main studies were identified by staff at the ONS using live birth registrations for births within two specific weeks: 2–8 January (pilot) and 4–10 March 2006 (main study). The same method of sampling was used as had been employed in 1995 to enable direct comparisons. Random samples of 400 women for the pilot survey and 4800 women for the main survey who were aged 16 years and over and who had delivered their baby in a 1-week period in England were selected. The sampling was stratified on the basis of births in different geographical areas (Government Office Regions). No subgroups were oversampled. The usable response rate was 60% for the pilot survey and 63% for the main survey. The samples included 229 women of BME born outside the UK. |
| 48 | To explore the healthcare experience of vulnerable pregnant migrant women. | London | Mixed methods: participants were contacted by phone (using a three-way interpreter call if appropriate) and interviewed using a pro forma questionnaire designed to determine their access to antenatal care; barriers to that access; and their experiences during pregnancy, labour and the immediate postnatal period. Further data was extracted from their records at the Doctors of the World (DOTW) clinic to see how they had accessed the clinic. | Not specified. | Phone survey. | Qualitative: thematic analysis. Quantitative: not clear. | Pregnant women who presented to the drop-in clinic of the DOTW in London were approached between January 2013 and June 2014. |

Continued

**Table 1** Continued

| Reference | Study aim | Region | Theory or framework | Methodology | Setting | Data analysis | Sample and mode of recruitment |
|---|---|---|---|---|---|---|---|
| 49 | To develop a reliable and valid questionnaire to evaluate satisfaction with maternity care in Sylheti-speaking Bangladeshi women. | London | Not specified. | Mixed methods: two-stage psychometric study. First, a Sylheti-language questionnaire regarding Bangladeshi women's experiences of maternity services was translated and culturally adapted from an English-language questionnaire using focus groups, in-depth interviews and iterative methods. Second, quantitative psychometric methods were used to field test and evaluate the acceptability, reliability and validity of this questionnaire. | Not specified. | Qualitative: thematic analysis. Quantitative: validity of an instrument. | Located at four hospitals providing maternity services in London, UK. Study participants included 242 women from the London Bangladeshi communities who were in the antenatal (at least 4 months pregnant) or postnatal phase (up to 6 months after delivery). The women spoke Sylheti, a language with no accepted written form. In stage one purposive samples of 40 women in the antenatal or postnatal phase participated, along with one convenience sample of six women in the antenatal phase and three consecutive samples of 60 women in the postnatal phase. In stage two, 135 women (main sample) completed the questionnaire 2 months after delivery (82% response rate), and 50 women (retest sample) from the main sample completed a second questionnaire 2 weeks later (96% response rate). |
| 50 | A Sure Start local programme had funded a Bangladeshi support worker to provide bilingual breastfeeding support to childbearing Bangladeshi women, many of whom were not fluent in English. This study aimed to conduct a short evaluation of the impact of this work on the uptake and duration of breastfeeding among these women. | Tower Hamlets | Not specified. | Mixed methods: the survey questionnaire included some open and closed questions about the women's intention to feed; their current feeding methods; the breastfeeding support and information they received antenatally, during the hospital stay and postnatally; overall views on the information and support received; and some demographic details. Eleven interviews were conducted by telephone in Sylheti (a dialect that has no written format), three in English and one in Urdu (using a female family member to translate). Interviews took between 15 and 30 min to complete. | Not specified (survey conducted by telephone). | Qualitative: content analysis of a questionnaire (open and closed questions). | The two midwives and the support worker had provided breastfeeding support in a 1-year period (September 2001 to August 2002). Of these, 80 women received help from the support worker alone. The majority of these 80 women were Bangladeshi. For the evaluation, 15 women were randomly selected from these 80 women. |
| 51 | To explore the perspectives of first-generation and second-generation women of Pakistani origin on maternity care and to make recommendations for culturally appropriate support and care from maternity services. | West Midlands | Retrospective Q method study. | Mixed methods: a retrospective Q methodology study of Pakistani women following childbirth. | Not specified. | Qualitative: Q methodology. | A purposive sampling strategy was used. Postnatal first- and second-generation Pakistani women were self-identified by their responses to information leaflets disseminated at local Children's Centres across an inner city in the West Midlands. |
| 52 | To evaluate a pilot mental health service for asylum-seeking mothers and babies. | UK (not clear) | Participatory action research framework. | Mixed methods: evaluation within a participatory action research framework. | Not specified. | Qualitative: thematic analysis. Quantitative: the CARE-Index. | An active outreach recruitment strategy was adopted by psychologists, who embedded themselves in a drop-in community group, the Merseyside Refugee & Asylum Seekers & Asylum Seekers Pre- & Postnatal Support Group. Participants were West African women who were asylum-seekers or refugee and who were either pregnant or had a young baby. They originated from The Gambia, Sierra Leone, Ivory Coast and Nigeria. All spoke English. Their ages ranged from 17 to 32 years, and all babies were under 6 months of age at the point of initial contact, with three babies not yet born. Attendance at the 21 therapeutic group sessions ranged between 4 and 12 mothers (with their babies). Seven mothers attended a significant proportion or all group sessions. An additional six mothers attended 1–4 group sessions. |

Continued

**Table 1** Continued

| Reference | Study aim | Region | Methodology | Theory or framework | Setting | Data analysis | Sample and mode of recruitment |
|---|---|---|---|---|---|---|---|
| 53 | To provide locally applicable data on the needs of Black and minority ethnic women in relation to their uptake of maternity and neonatal care provision by primary healthcare teams in Leeds. | Leeds | Mixed methods: questionnaires and focus groups. Interpreters were used when necessary for data collection. A questionnaire was translated into Urdu for some women. | Not specified. | Local community centres and in the participants' homes. | Qualitative: content analysis. Quantitative: survey (not clear). | A total of 97 questionnaires were completed, of which 50 were completed through informal links at community centres, schools and in women homes. The remaining 47 were completed while the researcher attended various antenatal clinics in the community. |
| 54 | To explore perinatal clinical indicators and experiences of postnatal care among European and Middle Eastern migrant women and to compare them with those of British women at a tertiary hospital in the North East of Scotland. | North East of Scotland | Mixed methods. Phase 1 of the research was a secondary analysis of routine data for 15 030 consecutive deliveries at Aberdeen Maternity Hospital. Phase two was a retrospective study of 26 European, Middle Eastern and British mothers in this hospital. After the women had given birth, verbal data was collected using face-to-face semistructured interviews. | Not clear. | Phase 2: 24 interviews were conducted in the homes of participants and two interviews at the university department. | Qualitative: thematic analysis. Quantitative: Phase 1 was a secondary analysis of routine data for 15 030 consecutive deliveries at Aberdeen Maternity Hospital. Phase 2 was a retrospective study of women. | Phase 1: The 15 030 deliveries included all births at Aberdeen Maternity Hospital over the financial years 2004–2008 in which maternal nationalities were identified and gestation was ≥24 weeks. Both singleton and multiple births were included. The clinical data was harvested from the Patient Administration System and the PROTOS maternity information system. In the case of women with multiple order births during the study, all births were included. Phase 2 of the research was a retrospective study of a few of the mothers who had given birth at this hospital. Eight European and five Middle Eastern women were semimatched with 13 British women. |
| 55 | To assess the mechanisms of support available to EM (ethnic minority) communities from community and voluntary sector organisations in relation to maternal and infant nutrition (a mapping exercise); to explore the experiences of the targeted client groups in seeking and receiving such support; and to identify gaps and opportunities to enhance support mechanisms and engagement with diverse EM communities. | Glasgow, Edinburgh, Aberdeen, Stirling, Fife, Dundee and Inverness. | Mixed methods: an online questionnaire survey of organisations working with EM communities, focus groups and telephone interviews with EM women. | Not specified. | Not specified. | Qualitative: thematic analysis. Quantitative: | The study identified 65 community organisations that potentially provided food and health services across EM communities in Scotland. In total, 37 organisations replied to the survey. Of those organisations, 15 indicated that they are providing services in the area of maternal and infant nutrition. A further 12 indicated that despite working with EM communities, they do not provide services in maternal and infant nutrition or healthy eating in general. An additional ten organisations confirmed by telephone that they were or had been working with EM women, but were unable to undertake the survey. The majority of interviewees for the focus groups and interviews were selected in response to a request sent by Black and Ethnic Minorities Infrastructure in Scotland (BEMIS) to community organisations. Snowball sampling was used to provide further contacts. In total, four focus groups were conducted with Polish, Roma, Czech and African mothers. In addition, six telephone interviews were conducted with Polish mothers. We focused on Polish mothers because they were the largest new ethnic group in Scotland since 2004. |
| 56 | To understand the nature of need in super-diverse areas and to examine the emergent challenges for effective maternity service delivery in an era of superdiversity. | West Midlands | Mixed methods: the study used a semistructured questionnaire and held narrative interviews of newcomer women. The findings were then triangulated with interviews of professionals who regularly worked with such women. | Not specified. | Not specified. | Qualitative: systematic thematic analysis. Quantitative: triangulation of findings. | Sampling was not described clearly. However, the study used a semistructured questionnaire that was designed in collaboration with maternity professionals and community researchers to explore the views and maternity experiences of newcomer women. Experienced multilingual female community researchers completed 82 of these questionnaires with interviewees in a range of different languages. Narrative interviews were also held with 13 women to further explore issues. The findings were triangulated with 18 interviews of professionals who regularly worked with migrant women. |
| 57 | To study the maternity care experiences of Somali refugee women in an area of West London. This article focused particularly on findings relating to the language barrier, which to a large degree underpinned or at least aggravated other problems the women experienced. | West London | Qualitative: case study. Six semistructured interviews and two focus groups (with six participants each). | Not specified. | Not specified. | Qualitative: thematic analysis. | Snowball sampling: 12 Somali women were selected from a larger survey involving 1400 women. |

Continued

**Table 1** Continued

| Reference | Study aim | Region | Methodology | Theory or framework | Setting | Data analysis | Sample and mode of recruitment |
|---|---|---|---|---|---|---|---|
| 58 | To undertake a qualitative study of the maternity experiences of 33 asylum-seekers. | London, Plymouth, Hastings, Brighton, Oxford, Manchester and King's Lynn. | Qualitative. | Not specified. | Home or a neutral location. | Qualitative: content analysis. | Convenience and snowball sampling of recent asylum-seekers. Based on semistructured interviews carried out in seven English cities. |
| 59 | To explore and synthesise the maternity care experiences of female asylum-seekers and refugees. | UK | Qualitative: multiple exploratory longitudinal case studies that used a series of interviews, photographs taken by the women, field notes and observational methods to contextualise data obtained during 2002 and 2003. | Theory of interactions and transformational educational theory. | Hospital settings or women's homes. | Qualitative: thematic analysis. | Women were approached if the status of 'asylum-seeker' or 'refugee' was written in the hospital notes taken at their booking appointment. Fourteen women were approached, but nine women declined to participate. Five women consented, but one woman was dispersed before 20 weeks gestation and therefore was not included in the study. Of the remaining four participating women, three were asylum-seekers and one was a refugee. The sampling technique was not clearly reported. |
| 60 | To identify key features of communication across antenatal care and whether they are evaluated positively or negatively by service users. | Central London | Qualitative: used six focus groups of 15 participants each and conducted 15 semistructured interviews. Non-English-speaking focus groups and interviews were conducted in standard Bengali, Sylheti or Somali. | Not specified. | Focus groups: hospitals and university meeting rooms. Semistructured interviews: various locations to suit the needs of the women. | Qualitative: thematic analysis. | The sampling technique was not clearly reported, but they recruited 30 pregnant women from diverse social and ethnic backgrounds affiliated with one NHS Trust (ie, hospital) in central London. Participants were recruited within this hospital, in eight community antenatal clinics situated in socially and ethnically diverse areas, via a community parenting group for Somali women, and via a Bengali Women's Health Project. Within the hospital, participants were recruited from the antenatal waiting room (which services low-risk and high-risk pregnancies), the ultrasound clinic and the glucose tolerance testing clinic. |
| 62 | To address the research question that postulates that immigrant women experience sensitive care through the use of an ethnically congruent interpreter and that such women prefer to meet health providers of the same ethnic and gender profile when in a multiethnic obstetrics care setting. | Greater London | Qualitative: in-depth individual and focus group interviews. Open-ended questions were presented by an obstetrician and an anthropologist. | Framework of naturalistic enquiry. | Not specified. | Qualitative: naturalistic inquiry. | Participants were recruited throughout Greater London between 2005 and 2006. Snowball sampling was used to recruit 36 immigrant Somali women, and another three were selected by a by purposive technique for a total of 39. A purposive technique was used to select further 11 Ghanaian women who had delivered at least one child within the British healthcare system and who were living within the study area at the time of data collection. |
| 61 | To study the relationships between Somali women and their Western obstetric care providers. The attitudes, perceptions, beliefs and experiences of both groups were explored in relation to caesarean sections, particularly to identify factors that might lead to adverse obstetric outcomes. | Greater London | Qualitative: in-depth individual and focus group interviews. | Framework of naturalistic enquiry, emic/etic model. | Not specified. | Qualitative: emic/ etic analysis. | Selected 39 Somali women by snowball sampling, 36 from the community and three purposively from a hospital. |
| 63 | To investigate women's experiences of dispersal in pregnancy and to explore the effects of dispersal on the health and maternity care of women asylum-seekers who were dispersed during pregnancy in the light of National Institute for Health and Care Excellence guidelines on antenatal, intrapartum and postnatal care. | London, South of England, Midlands and East of England, North West, North East and Wales. | Qualitative: interviews were conducted with 19 women who had been dispersed during their pregnancies and with one woman kept in an Initial Accommodation Centre under a new Home Office pregnancy and dispersal guidance issued in 2012. | Not specified. | Not specified. | Qualitative (not clear). | The sampling technique was not mentioned clearly. The women interviewed came from 14 different countries and had been dispersed or relocated to or within six regions of the UK. At the time of dispersal, 14 had been awaiting a decision on their asylum claim and six had been refused asylum. |

Continued

**Table 1** Continued

| Reference | Study aim | Region | Methodology | Theory or framework | Setting | Data analysis | Sample and mode of recruitment |
|---|---|---|---|---|---|---|---|
| 64 | To understand the multiple influences on behaviour and hence the risks to metabolic health of South Asian mothers and their unborn children, to theorise how these influences interact and build over time, and to inform the design of culturally congruent, multilevel interventions. | London boroughs, Tower Hamlets and Newham. | Qualitative: group story-sharing sessions and individual biographical life-narrative interviews. | Multilevel ecological models. | All but four interviews were in the participants' homes. | Qualitative: phenomenology. | The study recruited from diabetes and antenatal services in two deprived London boroughs 45 women of Bangladeshi, Indian, Sri Lankan or Pakistani origin aged 21–45 years with histories of diabetes in pregnancy. Overall, 17 women shared their experiences of diabetes, pregnancy and health services in group discussions, and 28 women gave individual narrative interviews (facilitated by multilingual researchers). All were audiotaped, translated and transcribed. |
| 65 | To gain an understanding of infant feeding practices among a group of UK-based refugee mothers. | Liverpool and Manchester | Qualitative: two focus group discussions and 15 semistructured interviews. | Not specified. | HCPs: private offices or clinics Refugee women: private rooms or discrete areas at the support venue (community centre or church hall). | Qualitative: thematic analysis. | The study purposively selected 30 refugee mothers from 19 countries who now resided in Liverpool or Manchester and were at least 6 months pregnant or had a child who had been born in the UK in the last 4 years. Of these 30, 19 were HIV negative and 11 were HIV positive. |
| 66 | To explore how Somali women with FGM experienced and perceived antenatal and intrapartum care in England. | Birmingham | Qualitative: a descriptive, exploratory study using face-to-face semistructured interviews that were audio-recorded. | Not specified. | Private room. | Qualitative: thematic analysis. | The study used convenience and snowball sampling of ten Somali women in Birmingham who had received antenatal care in England in the past 5 years. |
| 67 | To explore the maternity care experiences of pregnant asylum-seeking women in West Yorkshire to inform service development. | West Yorkshire | Qualitative: interpretative approach within the tradition of hermeneutic phenomenology. | Not specified. | Not specified. | Qualitative: interpretive approach with hermeneutic phenomenology analysis. | Purposive sampling was performed through the voluntary sector and a children's centre. In addition, word-of-mouth led to an element of snowball sampling. Six women were recruited. |
| 68 | To explore differences in infant thermal care beliefs between mothers of South Asian and white British origin in Bradford, UK. | Bradford District, West Yorkshire | Mixed methods: mothers were interviewed using a questionnaire with structured and unstructured questions. | Not specified. | The women chose the location of the interview. | Qualitative: thematic analysis. | A total of 102 mothers (51 South Asian and 51 white British) were recruited in Bradford District, West Yorkshire, UK. The inclusion criteria specified infants aged 13 months or less with a parent of South Asian or white British cultural origin who lived in the Bradford District. South Asia was defined as including the countries of Pakistan, India, Afghanistan, Sri Lanka and Nepal. Recruitment was aided by local community organisations, children's centres and community contacts. Urdu-speaking and Punjabi-speaking interpreters were requested and provided for 69 per cent of the first-generation South Asian mothers (n=26) in the sample. |
| 69 | To study the effectiveness of three linkworker and advocacy schemes that were designed to empower minority ethnic community users of maternity services. | Birmingham | Qualitative: focus group discussions, semistructured interviews and non-directive interviews. | Not specified. | Antenatal clinics in hospitals and health centres, community group settings and participants' homes. | Qualitative: not clear, thematic analysis? | Individual interviews were conducted with 66 Asian women who had received support from link-worker and advocacy services during their pregnancy and postnatally. Of these, 28 were from Birmingham, 13 from Leeds and 25 from Wandsworth-London. A semistructured interview guide was translated into five Asian languages: Hindi, Punjabi, Gujarati, Urdu and Tamil. The study also included ten focus groups made up of 60 women who had not used linkworker or advocacy services. All participants were recruited with the help of various minority ethnic women's groups and community organisations. Interpreters assisted 11 personal interviews with non-users from Vietnamese and Chinese backgrounds. |

Continued

**Table 1** Continued

| Reference | Study aim | Region | Methodology | Theory or framework | Setting | Data analysis | Sample and mode of recruitment |
|---|---|---|---|---|---|---|---|
| 70 | To study the maternity services experiences of Muslim parents in England. | UK: not specified | Qualitative: focus groups with Muslim mothers to explore their experiences of and views about maternity services; questionnaires with Muslim fathers; and interviews with health professionals | Not specified. | Not specified. | Qualitative: content analysis. | A mixed sample of 43 immigrants and non-immigrants were recruited via their project advisory groups. The focus groups were conducted in various locations around the UK, with two focus group discussions in a language other than English. A total of eight health professionals were interviewed: six midwives (two of whom worked for Sure Start programmes), a health visitor and a consultant obstetrician. |
| 71 | To explore the perceptions of pregnant asylum-seekers in relation to the provision of their maternity care while in emergency accommodation in the UK. | South East of England | Qualitative: an exploratory approach using unstructured interviews with five healthcare professionals and semistructured interviews with ten pregnant asylum-seekers. | Not specified. | Participants' emergency accommodations. | Qualitative: thematic analysis. | Purposive sampling of those providing maternity care for asylum-seekers produced a sample comprising two midwives (M1 and M2), one GP (GP), one hospital consultant (C) and one nurse (N), all based in south coast health centres and hospitals. A total of 15 pregnant asylum-seekers were approached to participate in the study. These women entered the UK through a south coast port over a 3-month period. Their countries of origin were Algeria, Congo, Angola, Nigeria, Somalia and Iraq, and they spoke French, Portuguese, Yoruba, Arabic and Kurdish. Translated information letters and consent forms were distributed to pregnant asylum-seekers via the Refugee Help Line, which also returned signed consent forms. This constitutes non-probability, purposive sampling. |
| 72 | To explore the meanings attributed by migrant Arab Muslim women to their experiences of childbirth in the UK. In particular, to explore migrant Arab Muslim women's experiences of maternity services in the UK; to examine the traditional childbearing beliefs and practices of Arab Muslim society; and to suggest ways to provide culturally sensitive care for this group of women. | UK: not specified | Qualitative: an interpretive ontological-phenomenological perspective informed by the philosophical tenets of Heidegger (1927/1962). | Heideggerian hermeneutic phenomenology. | All interviews were in the participants' homes except for one, which took place in a restaurant after 22:00 hours. | Qualitative: thematic analysis. | Purposive sampling produced eight Arab Muslim women who had migrated to one multicultural city in the Midlands. |
| 73 | To examine the health-seeking behaviours of Korean migrant women living in the UK. | London | Qualitative: 21 semistructured interviews. | Foucauldian approach. | Not clear. | Qualitative: not clear. | Women were recruited from New Malden via Korean community contacts. |
| 74 | To explore the experiences of obstetric care in Scotland among women who have undergone FGM. | Glasgow and Edinburgh | Qualitative: personal experiences of FGM and interviews. | Interpretivism paradigm and feminist perspective. | The Dignity Alert & Research Forum office or in the participant's home. | Qualitative: thematic analysis. | Convenience and purposive sampling resulted in a total number of seven women taking part in this study. All women were of African origin living in Scotland (three in Glasgow and four in Edinburgh). The inclusion criteria for the study were women who have undergone FGM and had experienced childbirth in Scotland. Three women were originally from The Gambia, two from Somalia, one from Ghana and one from Sudan. Six of them were Muslims and one was Christian. All women had undergone FGM in their countries of origin. Four women had been infibulated and the remaining three could not tell if they have had FGM type 2 or 3. |
| 75 | To gain a rich understanding of migrant Pakistani Muslim women's experiences of postnatal depression within motherhood; to inform clinical practice; and to suggest ways of improving supportive services for this group. | East London | Qualitative: interpretative phenomenology. | Interpretative phenomenological analysis (IPA) theory. | Not specified. | Qualitative: interpretative phenomenology. | Purposive sampling resulted in the recruitment of four migrant Pakistani Muslim women from London aged from 27 to 39. |

Continued

**Table 1** Continued

| Reference | Study aim | Region | Methodology | Theory or framework | Setting | Data analysis | Sample and mode of recruitment |
|---|---|---|---|---|---|---|---|
| 76 | To explore relationships between first-generation migrant Pakistani women and midwives in the South Wales region, focusing on the factors that contribute to these relationships and the ways that these factors might affect the women's experiences of care. | South Wales | Qualitative: a focused ethnography. | Symbolic interactionism. | Midwives: at lunch break or between clinics. Pakistani women: not clear. | Qualitative: thematic analysis. | Purposive sampling, through midwife gatekeepers, was selected for the initial recruitment of pregnant migrant Pakistani women: emails were sent to all midwives working with migrant women in South Wales. Snowballing was then used to recruit other midwives eligible for participation. Focused, non-participant observations of antenatal booking appointments took place in antenatal clinics across the local health board region over a period of 3–6 months. A total of seven midwives and 15 women were observed during these appointments, which lasted 20–60 min each. |
| 77 | To explore BME women's experiences of contemporary maternity care in all over England. | England | Qualitative data collected from a large cross-sectional survey using three open-ended questions that encouraged participants to articulate their experience of maternity care in their own words. | Not specified. | Not specified. | Qualitative: Thematic analysis. | A random sample of 4800 women was selected using Office for National Statistics birth registration records. The overall response rate was 63% but was only 3% from BME groups. A total of 368 women self-identified as coming from BME groups. Of those, 219 (60%) responded with open text and 132 (60%) were born outside the UK. |

PROTOS is a maternity care recording system.
HCP, healthcare professional.

appraised by two reviewers using tools from the Center for Evidence-Based Management (CEBMa).[34] We used Good Reporting of A Mixed Method study (GRAMM)[35] for the mixed-methods studies. Differences were resolved in our reflective team meetings. We also used high, medium and low as appraisal categories (discussed in table 2) This is approach is congruent with recent publications from the Cochrane Qualitative Research Group's Confidence in the Evidence from Reviews of Qualitative research (CERQUAL) publications and was previously used by in published studies by Higginbottom and colleagues.[79] Studies were classified in three into domains, high, medium and low, to enable a 'macro' evaluation.

▶ *High* was assigned to studies that used a rigorous and robust scientific approach that largely met all CEBMa benchmarks, perhaps equal to or exceeding 7 out of 10 for qualitative studies, 9 out of 12 for cross-sectional surveys or 5 out of 6 for mixed-methods research.

▶ *Medium* was assigned to studies that had some flaws but that did not seriously undermine the quality and scientific value of the research conducted, perhaps scoring 5 or 6 out of 10 for qualitative studies, 6 to 8 out of 12 for cross-sectional surveys or 4 out of 6 for mixed-methods research.

▶ *Low* was assigned to studies that had serious or fatal flaws and poor scientific value and scored below the numbers of benchmarks listed above for medium-level appraisals in each type of research.

The past decade has witnessed a growth in approaches to assessing quality and Popay *et al*[30] recommends evaluating not only the scientific quality of studies but also the 'richness' of studies, defined as 'the extent to which study findings provide in-depth explanatory insights that are transferable to other settings' (Popay *et al*, p230)[30] 'Thick' papers create or draw on theory to provide in-depth explanatory insights that can potentially be transferable to other contexts. By contrast, 'thin' papers provide a limited or superficial description and offer little opportunity for generalising. Each paper was assessed against the criteria as set out in Higginbottom *et al* (p5)[28 33] and categorised as either 'thick' or 'thin' (see table 2).

### Analysis and synthesis

Following construction of the preliminary themes, we produced code/narrative theme tables to demonstrate how the basic meaning units related to the theme. This involved utilising the codes produced in ATLAS.ti and aligning these to the manually extracted key findings (see figure 5). We reviewed all these processes in our reflective team meetings to ensure the rigour and robustness of our analytical steps. This iterative process is similar to the process of qualitative research and involved grouping the narrative findings into meaning units and social processes as they manifested in the maternity care experiences of immigrant women. Individual team members engaged in independent theming of tabular and coded data. We subsequently merged these individual perspectives to form the final harmonised themes representing a

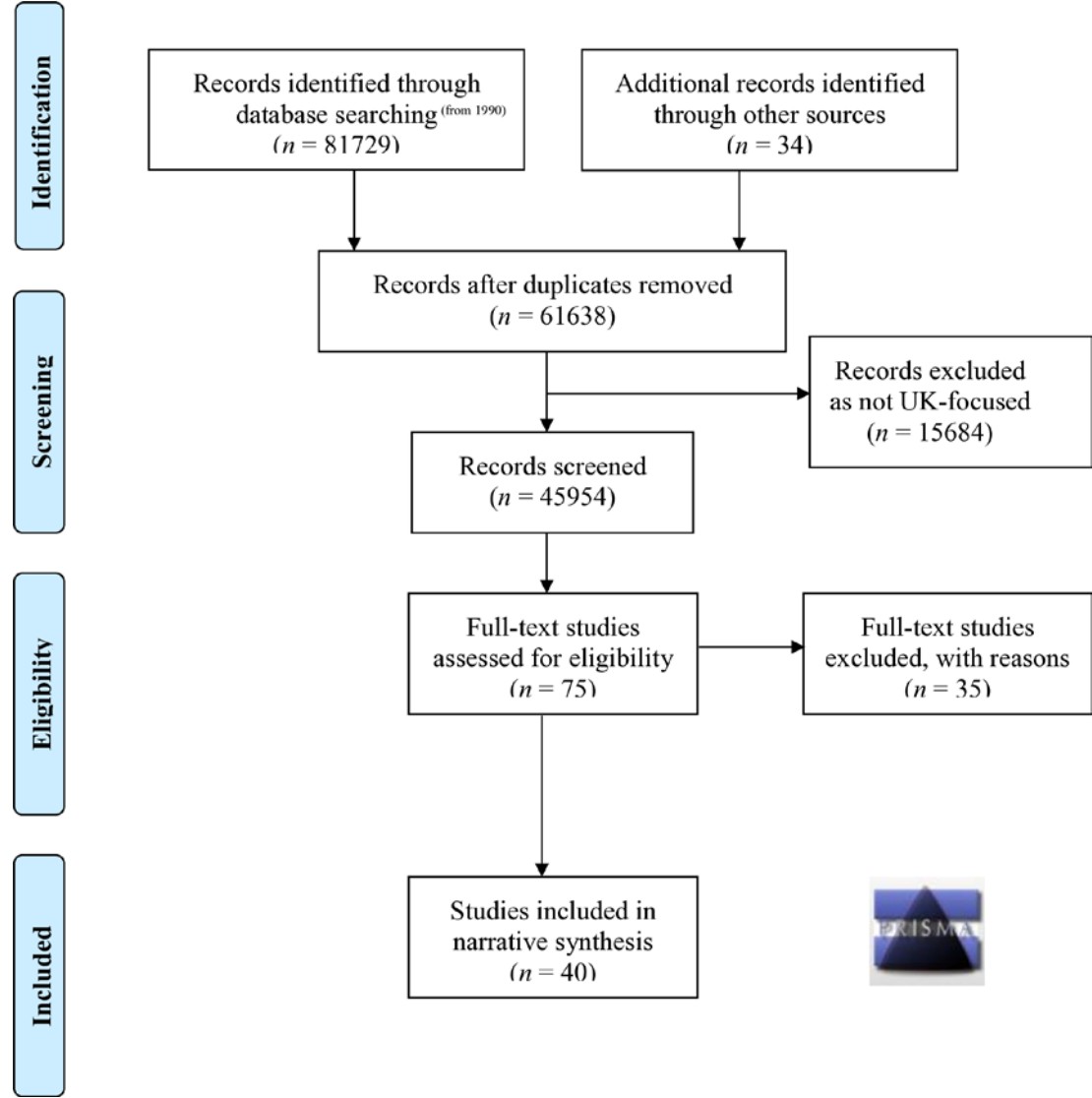

**Figure 2**  Preferred Reporting Items for Systematic Reviews and Meta-Analyses flow chart (from 1990).

'meta-inference' which is a term used in mixed methods research to describe merging of findings from the positivistic and the interpretative paradigms. Tashakorri and Teddlie (p101)[36] describe meta-inference as 'an overall conclusion, explanation of understanding developed from the integration of inferences obtained from the qualitative and quantitative strands'.

Following construction of the preliminary themes, we produced code/narrative theme tables to demonstrate how the basic meaning units related to the theme. Utilising the codes produced in ATLAS.ti and aligning these to the manually extracted key findings (see **figure 5**).

During the analytical processes we interrogated the data identifying using the concept suggested by Roper and Shapira.[37] We have constructed the themes in a policy directive fashion in terms of containing implicit indications in order to provide tangible guidance for policy and practice that might be developed into relevant strategies that benefit immigrant women and the NHS.

**Rigour, reflexivity and the quality of the synthesis**

Reflexivity in the review process requires a self-conscious and explicit acknowledgement of the impact of the researcher on the research processes, interpretations and research products. Reflexivity therefore demands acknowledgement of inherent power dimensions, hierarchies and prevailing ideologies that might shape and determine interpretations and the consequent knowledge production and research products. Gender, sexuality, professional socialisation, ethnocultural orientation and political lenses as these impact on social identities further coalescing to provide a specific perspective on any given phenomena. The review team members are imbued with a strong personal and professional commitment to the eradication of inequalities and allegiance to contemporary equality and diversity agendas. From a reflexive perspective, this is important given that immigration is global phenomena and the inherent vulnerability of some immigrant women.

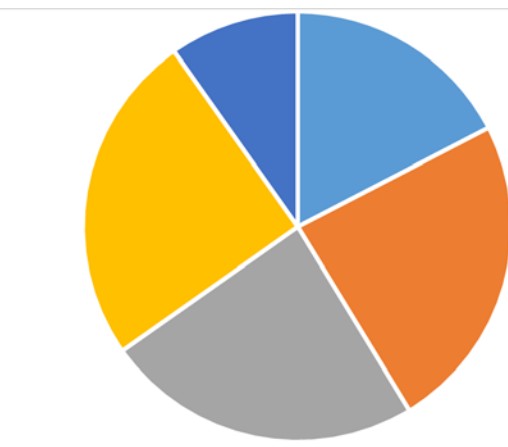

- Access and utilisation of maternity care services by immigrant women
- Maternity care relationships between immigrant women and healthcare professionals
- Communication challenges experienced by immigrant women in maternity care
- Organisations and legal entitlements and their impacts on the maternity care experiences of immigrant women
- Discrimination, racism, stereotyping, cultural sensitivity, inaction, and cultural clash in maternity care for immigrant women

**Figure 3** The total numbers of studies involved in each theme.

Reflexive analysis alerts us as researchers to emergent themes and informs the formal and systematic process of analysis, with reflexivity defined as:

> sensitivity to the ways in which the researcher's presence in the research setting has contributed to the data collected and their own a priori assumptions have shaped the data analysis (Murphy *et al*, p188)[38]

Our collaborative decisions required constant review and reading and, in some cases, reviewing the theme allocation and evidence to reach consensus. Therefore, we believe we achieved a nuanced and comprehensive approach. Higginbottom *et al* have successfully employed this review genre previously and have vast expertise in its usage.[39]

Within the published NS reviews, we have not given great attention to the issue of publication bias. However, we strove to eradicate any potential bias by undertaking a comprehensive and exhaustive literature review that included grey literature and follow-up emails with authors seeking greater clarity and explanation of opaque issues. A number of the included research studies were identified via *ProQuest* and *E-theses* and do not appear as publications in peer reviewed scientific journals.

We also held a national stakeholder event during which we presented our preliminary findings to a wide range of health professions (obstetrician, general practitioner and midwives), academics, voluntary and community workers. Possibly this approach may be considered contentious in the respect of systematic review, as attendees had no previous knowledge of the original included papers although they held deep topic knowledge. Notwithstanding this, we found broad support for our findings and facilitated groups work activities in order to

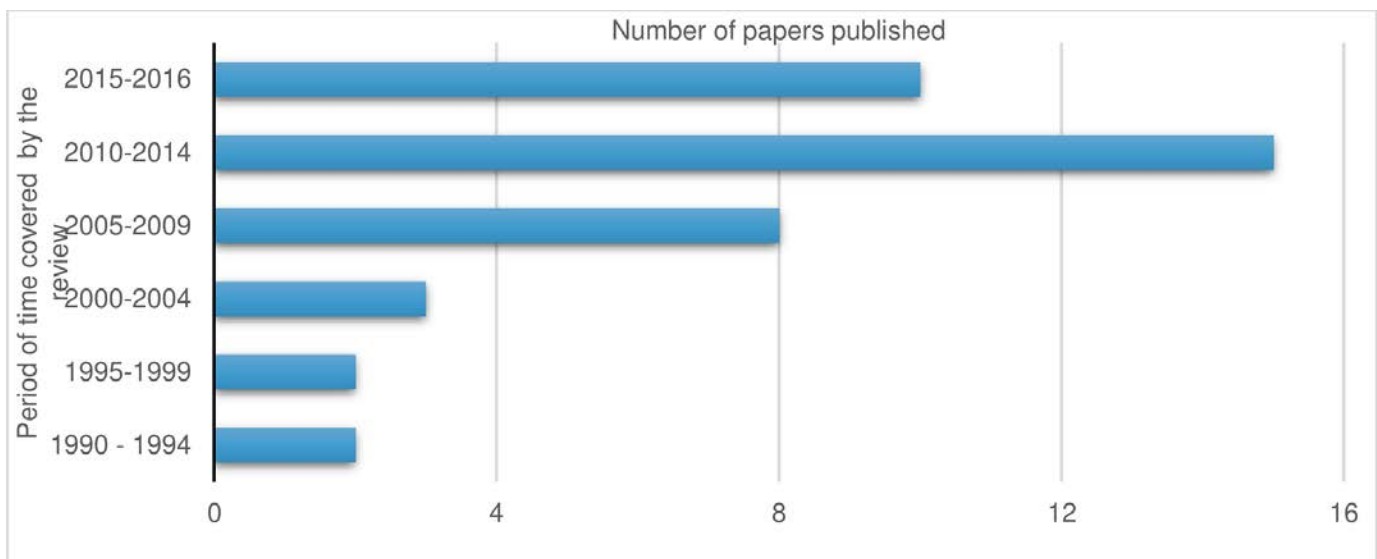

**Figure 4** The range of publication dates for the included studies (1990–2016).

**Table 2** Thick and thin criteria Higginbottom *et al*[33]

| Richness | Operational definition |
|---|---|
| Thick papers | ▸ Offer greater explanatory insights into the outcome of interest. <br> ▸ Provide a clear account of the process by which the findings were produced—including the sample, its selection and its size, with any limitations or bias noted—along with clear methods of analysis. <br> ▸ Present a developed and plausible interpretation of the analysis based on the data presented. |
| Thin papers | ▸ Offer only limited insights. <br> ▸ Lack a clear account of the process by which the findings were produced. <br> ▸ Present an underdeveloped and weak interpretation of the analysis based on the data presented. |

challenge our initial interpretations. These challenges resulted in the construction of *Theme 5: Discrimination, racism, stereotyping, cultural sensitivity, inaction and cultural clash in maternity care for immigrant women.* These focused activities collectively contribute to the confidence in the review findings, providing verification and validation of the themes.

We identified 40 research studies that met our inclusion criteria, and we extracted and synthesised key findings into five themes (see table 4) for the publications informing each theme.

### Methodological genres
#### Quantitative studies
We identified eight quantitative studies that all used a questionnaire for data collection.[40–47] These population-based studies and cohort surveys were all cross-sectional: none were longitudinal.

### Mixed-methods studies
We identified 10 mixed-methods studies that employed both qualitative and quantitative dimensions.[2 48–56] For example, Duff *et al*[49] reported a two-stage psychometric study in which focus groups and interviews were used in the first stage to develop a questionnaire for an ethnocultural group (Sylheti) In the second stage, quantitative methods were used to test and evaluate the acceptability, reliability and validity of the questionnaire. Other mixed-methods designs included (a) interviewing a small sample of the participants after collecting data from a large-scale survey; (b) conducting semistructured interviews with a small sample of participants based on quantitative data routinely collected from a large group of participants; and (c) using face-to-face, postal and online questionnaires to collect data. One of the studies used Q methodology which uses questionnaires with structured and unstructured questions.

### Qualitative studies
Of the 40 studies included in this review, we identified 22 as qualitative research studies employing a range of qualitative methodologies and approaches.[17 57–77] However, many of these studies did not specify a qualitative

**Table 3** Quality appraisal of the included studies

| Manual reference no | Quality as per the CEBMa tool | Relevance | Thick/thin |
|---|---|---|---|
| 1 | Low | High | Thin |
| 2 | Low | High | Thin |
| 3 | Low | High | Thin |
| 4 | Low | High | Thick |
| 5 | High | High | Thick |
| 6 | Med | High | Thick |
| 7 | Low | High | Thin |
| 8 | Low | High | Thin |
| 9 | Low | High | Thin |
| 10 | Med | High | Thin |
| 11 | Med | High | Thin |
| 12 | Med | High | Thin |
| 13 | Med | Low | Thin |
| 14 | Med | Med | Thin |
| 15 | High | High | Thin |
| 16 | High | High | Thin |
| 17 | Med | High | Thin |
| 18 | Med | Med | Thick |
| 19 | High | High | Thin |
| 20 | Med | High | Thick |
| 21 | High/medium | High | Thin |
| 22 | High | High | Thick |
| 23 | Med | Med | Thin |
| 24 | High | High | Thick |
| 25 | Med | High | Thin |
| 26 | High | High | Thick |
| 27 | Low | Med | Thin |
| 28 | Med | High | Thin |
| 29 | Low | High | Thin |
| 30 | Med | Med | Thin |
| 31 | High | High | Thick |
| 32 | Low | High | Thick |
| 33 | High | High | Thick |
| 34 | Low | High | Thin |
| 35 | High | High | Thick |
| 36 | Low | Low | Thin |
| 37 | Med | Med | Thin |
| 38 | Med | High | Thick |
| 39 | High | High | Thin |
| 40 | High | High | Thick |

CEBMa, Center for Evidence-Based Management.

methodological genre but instead employed a more generic qualitative approach and described only the data collection tools used. For example, some presented

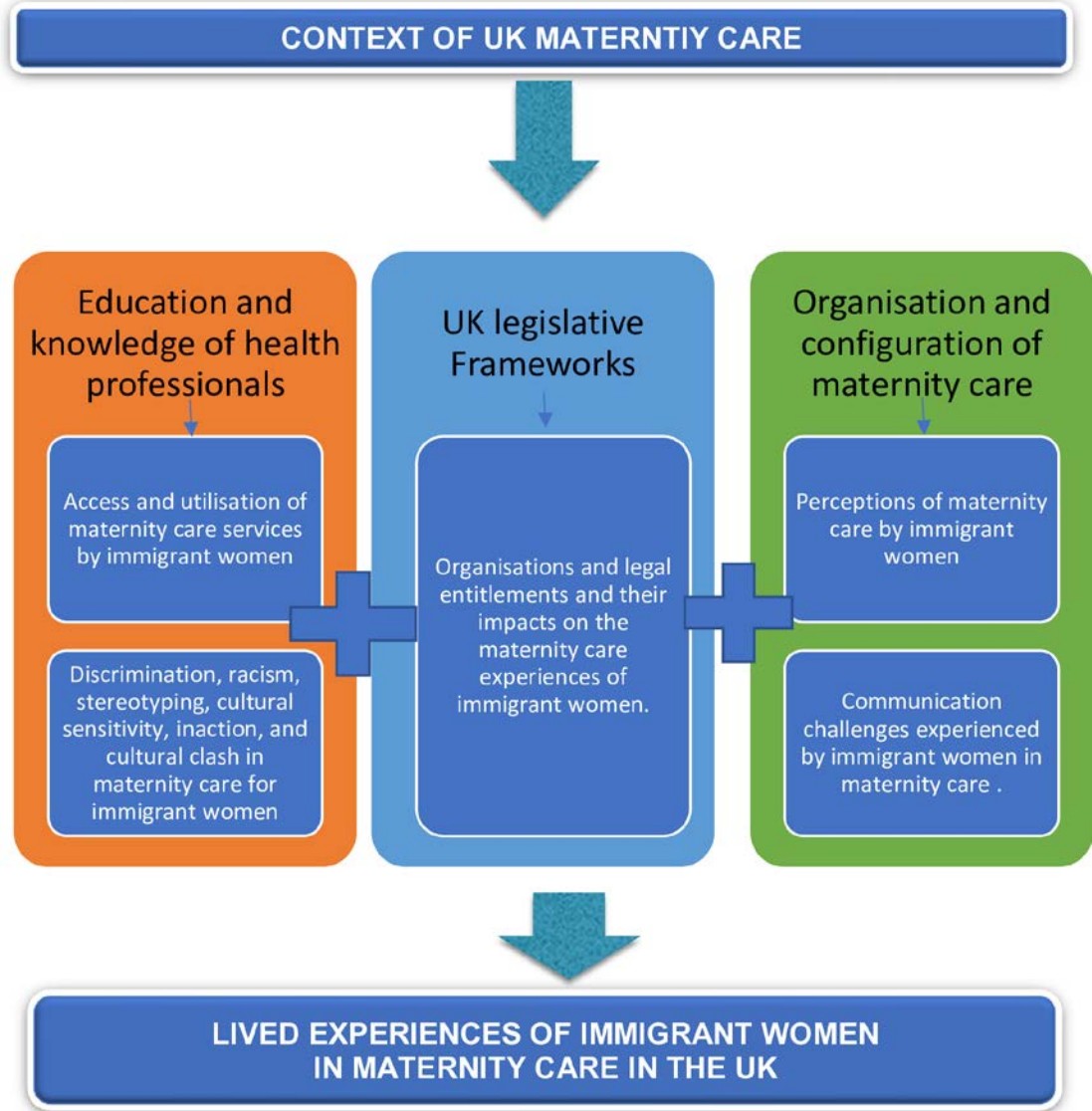

**Figure 5** Immigrant women's experiences of maternity care in the UK.

multiple longitudinal case studies of participants (asylum-seekers and refugees) about their maternity care experiences that included photographs taken by the participants, field notes and observations in addition to researcher interviews. Another example was a case study of an ethnocultural group, immigrant women of Somali origin, that used semistructured interviews and focus groups. Some studies used focus groups and interviews conducted in the language of the population group; for example, Bengali, Sylheti, Urdu and Arabic. Others used in-depth interviews, open-ended questions, group story-sharing sessions and individual biographical life-narrative interviews. In contrast, a few studies specified a qualitative interpretive approach that used hermeneutic phenomenology and focused ethnography.

### Studies focusing on specific ethnocultural groups
The chosen studies included participants from a wide range of ethnocultural groups that originated in diverse countries in different continents, including Asia (eg,

Bangladesh and Pakistan), Africa (eg, Somalia and Ghana) and Europe (eg, Poland). In some cases, the sample was drawn from a single ethnocultural group, such as Pakistani.[72] However, most of the studies were undertaken with mixed samples of immigrant women originating from different countries (eg, Somalia, Bangladesh and Eastern Europe) (see online supplementary file 6).

### Studies focusing on immigrant women without a clearly specified ethnocultural group
We identified 16 studies that used the term immigrant women generically and not clearly specify an ethnocultural group. In deciding to include these studies, we believed that legitimate proxies for immigrant status could be the specified use of an interpreter or the participants having countries of origin or birth outside the UK. Some studies reported immigrant women arriving from 14 different countries but did not specify the country of

**Table 4** Publications informing the themes

| Theme | 1<br>23 | 2<br>23 | 3<br>23 | 4<br>11 | 5<br>12 |
|---|---|---|---|---|---|
| Bawadi, HA (2009). *Migrant Arab Muslim women's experiences of childbirth in the UK.* | | X | X | X | X |
| Bazley Goodwin, LK (2016). *The midwife-woman relationship in a South Wales community: a focused ethnography of the experiences of midwives and migrant Pakistani women in early pregnancy.* | X | X | X | | |
| Hicks, C., & Hayes, L. (1991). *Link-workers in antenatal care: facilitators of equal opportunities in health provision or salves for the management conscience?* | X | | X | X | |
| Leeds Family Health (1992). *Research into the uptake of maternity services as provided by primary healthcare teams to women from black and minorities.* | X | X | X | | |
| Pershad, P., & Tyrrell, H. (1995). *Access to antenatal and postnatal services for Asian women living in East Pollokshields, Glasgow.* | X | X | X | X | |
| Warrier, S. (1996). *Consumer empowerment: a qualitative study of link-worker and advocacy services for non-English speaking users of maternity services.* | X | X | X | X | |
| Duff, L. A., Ahmed, L. B., & Lamping, D. L. (2002). *Evaluating satisfaction with maternity care in women from minority ethnic communities: development and validation of a Sylheti questionnaire.* | | | X | | X |
| Harper Busman, K., & McCourt, C. (2002). *Somali refugee women's experiences of maternity care in west London: a case study.* | X | X | X | X | X |
| Ali, N. (2004). *Experiences of maternity services: Muslim women's perspectives.* | X | X | X | X | X |
| MacLeish, J. (2005). *Maternity experiences of asylum seekers in England.* | X | X | X | X | X |
| Ahmed, S., Macfarlane, A., Naylor, J., & Hastings, J. (2006). *Evaluating bilingual peer support for breastfeeding in a local sure start.* | X | X | X | X | |
| Redshaw, et al. (2007). *Recorded delivery: a national survey of women's experience of maternity care 2006.* | | X | X | X | |
| Nabb, J. (2006). *Pregnant asylum-seekers: Perceptions of maternity service provision.* | X | X | | X | |
| Rowe, R. E., Magee, H., Quigley, M. A., Heron, P., Askham, J., & Brocklehurst, P. (2008). *Social and ethnic differences in attendance for antenatal care in England.* | X | | | | |
| Hawkins, S. S., Lamb, K., Cole, T. J., & Law, C. (2008). *Influence of moving to the UK on maternal health behaviours: Prospective cohort study.* | X | | X | | |
| Briscoe, L., & Lavender, T. (2009). *Exploring maternity care for asylum seekers and refugees.* | | | X | X | |
| Raine, R., Cartwright, M., Richens, Y., Mahamed, Z., & Smith, D. 2010). *A qualitative study of women's experiences of communication in antenatal care: identifying areas for action.* | X | X | X | | |
| Tucker, A., Ogutu, D., Yoong, W., Nauta, M., & Fakokunde, A. (2010). *The unbooked mother: a cohort study of maternal and foetal outcomes in a North London Hospital.* | X | | X | | |
| Lee, J-Y (2010). *'My body is Korean, but not my child's…': a Foucauldian approach to Korean migrant women's health-seeking behaviours in the UK.* | X | X | | | |
| Cross-Sudworth, F., Williams, A., & Herron-Marx, S. (2011). *Maternity services in multi-cultural Britain: using Q methodology to explore the views of first- and second-generation women of Pakistani origin.* | | X | | X | X |
| Essen, et al. (2011). *An anthropological analysis of the perspectives of Somali women in the West and their obstetric care providers on caesarean birth.* | | X | | | |
| Almalik, M. (2011). *A comparative evauation of postnatal care for migrant and UK-born women.* | X | X | X | X | |
| Binder, P., Borne, Y., Johnsdotter, S., & Essen, B. (2012a). *Conceptualising the prevention of adverse obstetric outcomes among immigrants using the 'three delays' framework in a high-income context.* | X | X | X | X | |
| O'Shaughnessy, R., Nelki, J., Chiumento, A., Hassan, A., & Rahman, A. (2012). *Sweet Mother: evaluation of a pilot mental health service for asylum-seeking mothers and babies.* | | | | X | |
| Binder, P., Johnsdotter, S., & Essen, B. (2012b). *Shared language is essential: communication in a multiethnic obstetric care setting.* | | X | X | | |
| Cresswell, J. A., Yu, G., Hatherall, B., Morris, J., Jamal, F., Harden, A., & Renton, A. (2013). *Predictors of the timing of initiation of antenatal care in an ethnically diverse urban cohort in the UK.* | X | | X | | |
| Jomeen, J., & Redshaw, M. (2013). *Ethnic minority women's experience of maternity services in England.* | | X | X | | X |

Continued

**Table 4**  Continued

| Theme | 1 | 2 | 3 | 4 | 5 |
|---|---|---|---|---|---|
| | 23 | 23 | 23 | 11 | 12 |
| BEMIS Scotland (2013). *A comparative evaluation of postnatal care for migrant and UK-born women.* | X | X | X | | |
| Baldeh, F. (2013). *Obstetric Care in Scotland: the experience of women who have undergone Female Genital Mutilation (FGM).* | | X | X | X | |
| Feldman, R. (2014). *When maternity doesn't matter: Dispersing pregnant women seeking asylum.* | X | | | X | |
| Gorman, D. R., Katikireddi, S. V., Morris, C., Chalmers, J. W. T., Sim, J., Szamotulska, K., … & Hughes, R. G. (2014). *Ethnic variation in maternity care: a comparison of Polish and Scottish women delivering in Scotland 2004–2009.* | X | | | | |
| Greenhalgh, T., Clinch, M., Afsar, N., Choudhury, Y., Sudra, R., Campbell-Richards, D., & Finer, S. (2015). *Socio-cultural influences on the behaviour of South Asian women with diabetes in pregnancy: Qualitative study using a multi-level theoretical approach.* | X | | | | |
| Phillimore, J. (2015). *Delivering maternity services in an era of superdiversity: The challenges of novelty and newness.* | X | X | X | X | X |
| Lamba, R. (2015). *A Qualitative Study Exploring Migrant Pakistani-Muslim Women's Lived Experiences and Understanding of Postnatal Depression.* | | X | X | | |
| Shortall, C., *et al.* (2015). *Experiences of Pregnant Migrant Women receiving Ante/Peri and Postnatal Care in the UK: A Doctors of the World Report on the Experiences of attendees at their London Drop-In Clinic.* | X | | X | X | |
| Moxey, J. M. & L. L. Jones (2016). *A qualitative study exploring how Somali women exposed to female genital mutilation experience and perceive antenatal and intrapartum care in England.* | X | X | X | X | |
| de Chavez, A. C., Ball, H. L., & Ward-Platt, M. (2016). *Bi-ethnic infant thermal care beliefs in Bradford, UK.* | | | X | | |
| Hufton, E., & Raven, J. (2016). *Exploring the infant feeding practices of immigrant women in the North West of England: A case study of asylum seekers and refugees in Liverpool and Manchester.* | | X | | X | |
| Phillimore, J. (2016). *Migrant maternity in an era of superdiversity: New migrants' access to, and experience of, antenatal care in the West Midlands, UK.* | X | | | X | |
| Lephard, E., & Hait.h-Cooper, M. (2016). *Pregnant and seeking asylum: Exploring women's experiences from booking to baby'.* | X | X | X | X | X |

birth. Withoutclearly specified ethnic group, these studies were still included.

### Theme 1: access and utilisation of maternity care services by immigrant women

Late booking emerged as an important dimension in this theme with immigrant women study participants often booking and accessing antenatal care later than the recommended timeframe of during the first 10 weeks of pregnancy. This delayed utilisation was found to be multifactorial in nature with influencing factors including the effects of limited English language proficiency, immigration status, lack of awareness of the services, lack of understanding of the purpose of the services, income barriers, the presence of female genital mutilation (FGM), factors associated with differences between the maternity care systems of their countries of origin and the UK, arrival in the UK late in the pregnancy, frequent relocations after arrival, the poor reputations of antenatal services in specific communities and perceptions of regarding antenatal care as a facet of medicalisation of childbirth. The range of factors affecting the access and utilisation of postnatal services were similar to those reported for antenatal services.

### Theme 2: maternity care relationships between immigrant women and HCPs

Our included studies identified the perception of service users in this group and their interactions and therapeutic encounters with HCPs as significant in understanding access, utilisation, outcomes and the quality of their maternity care experience.

Included studies identified both positive and negative perceptions of study participants regarding the ways HCPs delivered maternity care services were both positive and negative. A number of studies illustrated positive relationships between HCPs and immigrant women with the HCPs described as caring, respecting confidentiality and communicating openly in meeting their medical as well as emotional, psychological and social needs. Conversely, some studies provided evidence of negative relationships between participants and HCPs, with HCPs described from the perspective of immigrant women as being rude, discriminatory or insensitive to the cultural and social needs of the women. The end result of these negative encounters was that these women tended to avoid accessing utilising maternity care services consistently.

### Theme 3: communication challenges experienced by immigrant women in maternity care

It is axiomatic that limited English language fluency presents verbal communication challenges between HCP and their patients, families and carers. Moreover, this is compounded when HCPs use complex medical or professional language that is difficult to comprehend. Nonverbal communication is culturally defined and challenges can occur through misunderstandings of facial expressions, gestures or pictorial representations. Poor communications result as illustrated in our included studies in limited awareness of available services in addition to miscommunication with HCPs. Study participants often expressed challenges in accessing services, failed to understand procedures and their outcomes and were constrained in their ability to articulate their health or maternity needs to healthcare providers and disempowered in respect of their involvement in decision-making. They therefore sometimes gave consent for clinical procedures without fully comprehending the risks and benefits, and did not always understand advice on baby care. Studies also identified communication as not reciprocal with HCPs often misunderstanding participants. These issues of communication were described as leading to feelings of isolation, fear and a perception of being ignored.

### Theme 4: organisation and legal entitlements and their impacts on the maternity care experiences of immigrant women

The study participants in our included studies had mixed experiences with the maternity care services in the UK. Positive and commendable experiences included feeling safe in giving birth at hospital rather than at home, being able to register a complaint if poor healthcare was received, being close to a hospital facility, not being denied access to a maternity service, and having good experiences with postnatal care. Conversely, negative experiences included lack of continuity (eg, not being able to see same maternity care providers each time) and being unaware of the configuration of maternity services work that limited appropriate use. Participants in our included studies found services bureaucratic and perceived within the UK maternity care model as having a propensity towards medical/obstetric intervention and lower segment caesarean section births.

The legal status of an immigrant women in the UK has a profound influence on their on their access to maternity care. Women without entitlement to free maternity care services in the UK were deterred from accessing timely antenatal care by the costs and by the confidentiality of their legal status. Moreover, some women arrived in the UK during the final phase of their pregnancies that resulted in interruptions in the care process, loss of their social networks, reduced control over their lives, increased mental stress and increased vulnerability to domestic violence.

Positive experiences included receiving information from their midwives on the benefits of breastfeeding together with demonstrations on how to position the baby. Negative experiences included poor support from hospital staff on how to breastfeed their babies consequently these reported experiences are mixed.

### Theme 5: cultural sensitivity, inaction and cultural clash in maternity care for immigrant women

Inequalities in access, navigation, utilisation and the subsequent maternity care outcomes are influenced by discrimination and cultural insensitivity in maternity care services according to the perspectives of women in several included studies. Although discrimination is often subtle and difficult to identify, direct and overt discrimination was reported in some studies.

Specifically, study participants of Muslim faiths challenged assumptions held by HCPs, including those held regarding Muslim food practices and that their partners or husbands should help the women during labour. Moreover, HCPs were reported in some studies to lack cultural sensitivity and cultural understanding. For example, these women did not optimally benefit from antenatal classes facilitated by a non-Muslim educator who had no knowledge of the relationships of Muslim culture to maternity.

Furthermore, Muslim participants often expressed dissatisfaction with antenatal classes having a gender mix, which contravened religious edicts. Studies illustrated that some women of Muslim faith also regarded their cultural and religious needs were not met, and they felt that the staff lacked insight, knowledge and understanding of FGM.

Evidence from our included studies suggests some immigrant women perceived that the staff did not treat them with respect or attended fully to their healthcare needs, and they felt devalued, unsupported and fearful while receiving maternity care. Our findings also identified instances of cultural clash and conflicting advice during pregnancy and maternity care, mostly resulting from differences between the cultural practices and medical systems of the home countries of the immigrant women and those in the UK. In a few cases, however, midwives were happy to meet the cultural and religious needs of the study participants in our included studies in both antenatal and postnatal settings which is a positive finding.

We conceptualise the findings graphically in figure 5.

### Patient and public involvement

The systematic review questions were developed in consultation with our project advisory group including service users' priorities experience and preferences. This systematic review did not include empirical research therefore there were no human participants.

## DISCUSSION AND CONCLUSIONS

The UK is in a period of superdiversity, defined as being 'distinguished by a dynamic interplay of variables among an increased number of new, small and scattered, multiple-origin, transnationally connected,

socio-economically differentiated and legally stratified immigrants'. (Vertovec, p1024).[1] Responding to this level of diversity is challenging for UK maternity care health services and may require the development of new and innovative strategies.

The experiences of immigrant women in accessing, navigating and utilising maternity care services in the UK are both positive and negative. In order to enhance services, it is essential that strategies are developed to overcome the negative experiences reported. The experience of maternity care services is multifactorial in nature with a number of issues appearing to coalesce to determine the poorer experience reported by some immigrant women. Important factors identified by the review included a lack of language support, cultural insensitivity, discrimination, poor relationships between immigrant women and HCPs, and a lack of legal entitlements and guidelines on the provision of welfare support and maternity care to immigrants.

### Implications of findings and recommendations for maternity care policy, practice and service delivery

Inequitable access appeared to be a consequence of the immigration and legal status of asylum-seeking women which has a profound impact on healthcare experiences and consequently health, and was also influenced by language fluency. We concluded that addressing language barriers and ensuring culturally sensitive care are essential elements of providing optimal maternal care for immigrant women. The issue of confidentiality may be compromised by having known interpreters in small communities. One solution may be the setting up of a national-level website offering standard information on maternity care and the option of translation in a wide range of languages. Additionally, the identification of best language practices should be identified in order to improve the current language service model.

The knowledge, understandings and attitudes of maternity care healthcare providers is a critical determinant of care. Ethno-culturally based stereotypes, racism, judgemental views and direct and indirect discrimination require eradication requires challenging discrimination and racism at all levels: individual, institutional, clinical and societal Interventions to improve maternity care for immigrant women are scant, and formal evaluations of these interventions were largely absent. Increasing the social capital available to immigrant, health literacy and advocacy resources may empower women to access and use maternity care services appropriately.

Maternity care staff require a greater level of mandated education to have better cultural awareness of needs of diverse client groups including newcomers to the UK. Our findings highlight the importance of demonstrating compassion, empathy and warmth in their relationships with these women to reinforce positive attitudes among immigrant women.

It is contingent on maternity care providers to value diversity among service users and to offer individualised and culturally congruent care. One way to achieve this goal would be through birth plans that can be jointly agreed and discussed in advance by the maternity care staff and recently arrived newcomers and immigrant women. Maternity care staff should seek to empower immigrant women by providing comprehensible information and better education concerning the configuration of the maternity system in the UK, conveying accurate information about care delivery. Central to these suggestions may be to enable volunteer and third-sector organisations to work as links between the statutory maternity services and immigrant women. We found evidence (though not scientifically evaluated) of such links in our national networking event.

Representatives of immigration control agencies may feel obligated to adheres to immigrant rules and consider the maternity care needs of immigrant women's and baby's health as a secondary issue. The policy context regarding data protection and sharing information with the Home Office about the immigrant status of women was at issue as well, especially since variabilities have been seen in the policies for sharing this information. The results suggest that the legal and policy context is important in addressing the maternity care needs of immigrant women.

It would seem imperative, as reflected in current policy directives, to adopt a universal of aim of achieving optimal maternity care for all and not just for immigrant women. However, maternity care services should strive to give more information to immigrant women about their rights to care, the availability and configuration of maternity services, and how to navigate maternity care systems. The child in utero of an immigrant is a future UK citizen and optimising maternity care is a dimension of securing the future health of the nation. In a period of super diversity is incumbent on health professional to have an awareness of immigrant women's legal rights and perhaps education on this topic should be mandated for maternity HCPs. Continuity in maternity caregivers and compulsory provision of interpreters would also help to improve the experiences of these women.

Decision-makers and healthcare leaders should address the findings at a strategic level. A focus on diversity, equality and the needs of immigrant women could reasonably be embedded in the role and responsibility of 'Board level Maternity Champion 'and of 'Maternity Clinical Networks'. Maternity service providers could consider the appointment of one obstetrician and one midwife jointly responsible for championing maternity care provision to immigrant women in their organisation. As these dimensions feature within the 'Bespoke Maternity Safety Improvement Plan',[78] key areas of action include:

► Focus on learning and best practice: issues of equality and diversity should be featured in the Saving Babies' Lives care bundle for use by maternity commissioners and providers.
► Focus on multiprofessional team working: continuous personal and professional training.

► Focus on data: greater focus on ethnicity and immigration within the Maternity Services Dataset and other key data sets.

► Focus on innovation: create space for accelerated improvement and innovation at local level.

## Gaps in the evidence

Some locally developed and locally based interventions to address inequalities in access and quality in maternity care for immigrant women were described during the final feedback meeting. However, there are very few interventions to address these issues in the published literature and their effectiveness has not been evaluated robustly. None of the interventions had also included economic evaluation of the intervention. Studies of the usual 6 weeks postnatal checks by a general practitioner were not identified nor studies that focused on the intrapartum period. As mentioned earlier, we found few studies that focused on immigrant women with 'white ethnicity' in our review time period, for example, women of Eastern European origin.

## Strengths and limitations

► We were challenged and constrained by the lack of consistency in describing immigrant population sin the published literature. There exists a great deal of variation and no unified approach within the UK literature.

► Immigration is an international phenomenon, and this review increases understanding of how immigrant women navigate maternity services in the UK,

► The review systematically maps our positive and negative aspects of maternity care provision as experienced by immigrant.

► The review provides strategic policy-level direction for enhancement of maternity care services.

► The review does not address the experiences of maternity care for second-generation women (eg, women of black and minority origin born in the UK) nor does it consider refugee and asylum seeking women as a separate group.

## Implications for future research

More research is required into how the term 'immigrant' is used, and the changes in its use over time that may affect immigrant women's care. At present, the term is used very broadly and simplistically which masks its inherent heterogeneity. Furthermore, more research is required to understand how the intersections of particular characteristics—such as gender, education status, time in the UK, immigration status, wealth and country of origin—may influence or alter the experiences of these women in their maternity. Research is also required that focuses on developing and evaluating specific interventions to improve maternity care for immigrant women.

**Acknowledgements** We gratefully acknowledge valuable input from the members of our Project Advisory Group. Their input has been very helpful in making possible the successful completion and the high quality of this review. The following people kindly consented to be members: Jim Thornton, Professor of Obstetrics and Gynaecology, Faculty of Medicine and Health Sciences, University of Nottingham, Dr Caroline Mitchell, General Practitioner/Senior Clinical Lecturer, Clinical Academic Training Programme Lead, Academic Unit of Primary Medical Care (AUPMC), University of Sheffield. Dr Jane Mischenko, Commissioning Lead: Children and Maternity Services, NHS Leeds, Carol McCormack, Specialist Midwife, NUH Trust. We also thank following immigrant women for their input in the conceptualisation of this review: Ms Valentine Nkoyo, Director of Mojatu, Nottingham Kinsi Clarke, Nottingham Refugee Forum

**Contributors** GH (Professor, School of |Health Sciences) was principal investigator. Initiated the project and oversaw all stages. Led the interpretation/synthesis phases and draft-ed the manuscript. BH (Senior Research Fellow) contributed to all stages of the review. Led the data extraction, coding, and quality appraisal and contributed to the manuscript. CE (Associate Professor of Nursing, Director of the Centre for Evidence Based Health Care) contributed to all stages of the review, provided expert methodological advice, acted as second reviewer for quality appraisal and development of the synthesis. She contributed to the review of the final version of the manuscript. MM (Professor Emeritus King's College London) contributed to all stages of the review, provided expert methodological advice, acted as second reviewer for quality appraisal and development of the synthesis. She contributed to the review of the manuscript. KB (Retired Director of Midwifery, University of Leeds) contributed to all stages of the review, provided clinical and policy perspectives, contributed to formulation of the implications and recommendation in the manuscript. JE (Information Specialist) designed the literature search strategy, advised the team on all aspects of information retrieval, undertook the main database searches and contributed to the development of the manuscript.

**Funding** This project was funded by the UK National Institute for Health Research (NIHR) Health Services and Delivery Research Programme (Grant No. HS&DR-15/55/03). Along with this funding, NIHR also contributed by peer reviewing the funding proposal.

**Competing interests** None declared.

**Patient consent for publication** Not required.

**Provenance and peer review** Not commissioned; externally peer reviewed.

**Data availability statement** Data are available upon reasonable request.

**ORCID iDs**
Gina Marie Awoko Higginbottom http://orcid.org/0000-0001-8851-0985
Catrin Evans http://orcid.org/0000-0002-5338-2191

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
