## [Reviewer comments · BMJ Open]

ARTICLE DETAILS

TITLE (PROVISIONAL)	Experience of and access to maternity care in the United Kingdom (UK) by immigrant women: a narrative synthesis systematic review
AUTHORS	Higginbottom, Gina; Evans, Catrin; Morgan, Myfanwy; Bharj, Kuldip; Eldridge, Jeanette; Hussain, Basharat

VERSION 1 – REVIEW

REVIEWER	Prof Nusrat Husain University of Manchester
REVIEW RETURNED	12-Mar-2019

GENERAL COMMENTS	Immigrant women's experience of and access to maternity care in the United Kingdom (UK): a narrative synthesis systematic review Immigrant women experience numerous changes which may lead to psychological conflicts, social disintegration and decline in their physical and mental health. Considering the evidence around immigrant populations being more vulnerable to develop both physical (e.g., cardiovascular disease, diabetes, cancer, infectious diseases) and mental illnesses (e.g., depression, psychosis, post-traumatic stress) and poorer health outcomes particularly in relation to reproductive health, it is very important to understand the experiences this high risk population of using local maternity care services. Abstract: In methods, authors should say briefly about the inclusion criteria for the studies included in this SR, if not all, may be worth adding which type of studies were included (both quantitative and qualitative). In results most information is from qualitative studies, though they also included quantitative and mixed methods, I suggest authors briefly state some of the salient findings from quantitative studies also. Introduction: The authors started this section with a very interesting and important concept of super-diversity. They also highlighted the difficulties immigrant women face with regard to access and utilization of health services. On page number 2, paragraph 2 line 34 authors talk about life threatening incidents, it may be worth adding what these incidents are on the same page, references are missing on quite a few places such as line no. 7, 31, and 41. On page number 4 example given in line 20 is not clear and it may help to reword. The authors mention that they also categorized foreign students as immigrants which is a different group. Need to check paragraph starting from line 40 on this page as sentences are incomplete. In "aim and rationale" section, point "b" is difficult to understand. Can we rephrase it as "women perception about availability of services and their experiences of accessing these services". Figure 1, does not add much to what is already mentioned on page 5 line3-8.
--

	Methods: In this section authors mentioned that they used Popay's approach to narrative synthesis (NS). It is not clear how this approach is different to traditional NS methods. Authors mentioned that they were interested to include studies testing different interventions; it is not clear why they did not include RCTs. On page 8, authors mention criteria for high, medium and low quality studies but I was not able to find to find how many studies qualified each of these criteria. There is no information on how the quantitative data from cross sectional studies and quantitative section of mixed method studies was taken care of. On page 10 there is lot of detail under the heading "rigor, reflexivity and quality of synthesis" it will be good if it can be a bit more focused. Results and discussion: Over all this is a well written section with interesting results though things can be a bit clearer if the authors can tell us what results are from quantitative studies and what is from qualitative. There are a lot of typos throughout the manuscript. All the above is to improve the quality of the manuscript in my opinion this paper will be of interest to the readers of BMJ open.
--	--

REVIEWER	Christine McCourt City, University of London, England
REVIEW RETURNED	01-Apr-2019

GENERAL COMMENTS	General comments There were numerous typographical errors, incomplete or oddly edited sentences that impaired the readability of the article and the expression felt a bit stilted at times. e.g. lines 7-8 'That is a The UK is in a period of superdiversity defined as a "distinguished by a... The expression and grammar are at times poor; e.g. 'this defers from the former in definition in that the term migrant is conferred on the on the basis of nationality. Meaning all applicants that hold nationality other than the UK are considered migrants. However, the situation is dynamic in that the nationality of a person is may also to change over time and in some individuals may acquire dual-citizenship involving several nation states (lines 17-22). There are frequent other such examples, so the whole article needs considerable editing to improve the grammar, use of punctuation and the expression and proof read to ensure that sentences are actually complete and make sense. These are simply too time consuming for me to list as there are so many. I have to say that I was very surprised that the authors had submitted such a poorly prepared manuscript, which looked more like an earlier draft than a completed submission. Abstract The abstract was fairly clear, although the expression at a few points could have been improved. While you state 'We found few interventions that had focused on improving maternity care for these women and the effectiveness of existing interventions have not been rigorously evaluated' this is referring to specific interventions, so you do need to take into account that other interventions which are less specific may also have benefits for this group of women. This is a challenge of doing this kind of review and can be picked up in the article discussion.
--

	Also, I wasn't clear what 'We contacted stakeholders with expertise' referred to. Contacted for what purpose? To identify other studies? To comment on the analysis? ?? and expertise in what? Strengths and limitations highlights The following sentence seems incomplete: 'The review systematically maps our positive and negative aspects of maternity care provision as experienced by immigrant' I wasn't terribly convinced by these highlights as the review's scope, as noted under abstract, was to look at 'research that focused on access and interventions to improve maternity care' so was not quite as broad as this section seems to suggest. Also, the review question as stated in lines 46/47 is even narrower. Introduction The authors discuss various reviews which have been conducted, arguing that none covers the same topic. However, they missed the following published review: Small R, Roth C, Raval M, Shafiei T, Korfker D, Heaman M, McCourt C, Gagnon A. (2014) Immigrant and non-immigrant women's experiences of maternity care: a systematic and comparative review of studies in five countries. BMC Pregnancy and Childbirth 2014, 14:152 doi:10.1186/1471-2393-14-152 That was published in 2014 so an update could be the alternative justification, in addition to the specific focus of this review question on identifying interventions to improve access, but the authors need to amend the background to reflect the existence of a prior recent review on the subject. The definition of concepts and the account of the synthesis approach and use of Gulliford's theory were all fine, although shot through with grammatical and typographical errors. However, I found the statement of aims and research question rather muddled and confusing. A key issue was that the question stated is quite specifically focused on identifying interventions to improve access; however, this is not really reflected in the study title and abstract and only partly so in the objectives. Methods On pages 8-9 a section of text from 'Data Extraction and assessment of relevance' is repeated word for word in analysis and synthesis. Also, I didn't really understand how the data extraction table described facilitated a narrative approach. It just sounds like a standard study summary table. (The Roper and Shapira concept seemed to make more sense for that.) The way in which the critical appraisal was used was not very clear. Were low quality studies excluded? How did you relate the low/med/high classification to the thick/thin one? At what stage and how did you integrate the CA in the analysis?
--	---

	On page 11, lines 3-4 you say, 'Individual team members engaged in independent theming of tabular and coded data.' This is good but can you clarify at what stages and to what extent? All? A sample? How many people? Search findings The account of studies identified was a little incongruous in that it listed quantitative studies and mixed methods studies, but not the number/type of qualitative studies. Findings – analysis The findings section felt very descriptive and rather like a list of issues rather than analysis. The themes are all logical and reflect previous studies well but the opportunity to learn something new I felt had been missed. For example, contrasting experiences of care are noted with no attempt to identify or explore what kind of factors such as service model or design are associated with more positive or negative experiences of care encounters. Also, I couldn't see anything which attempted to answer the stated research question – i.e. interventions to improve access. Instead, what we get is a rather undifferentiated descriptive picture which doesn't really add to what is already known. Discussion/conclusions This section felt rather like a list of 'shoulds' all of which are fair enough and have already been stated in other studies, but surely the issue is to analyse in more depth how these imperatives, which have been stated before, might be achieved? I would have thought the research question might have helped to address this but it seems to be a gap in the review. The suggestion regarding birth plans is questionable as the evidence for the impact of these is very poor. The statement regarding interpreting and cultural competence of professionals is also not new, so again, the question might be better posed in relation to how this might be improved? Summary view/recommendation This is a weak article which is not suitable for publication. It doesn't address the actual stated review question and does not appear to add anything substantial to knowledge on the topic.
--	---

VERSION 1 – AUTHOR RESPONSE

Review Comment	Author Response
Reviewer: 1	

Review Comment	Author Response
In methods, authors should say briefly about the inclusion criteria for the studies included in this SR	
may be worth adding which type of studies were included (both quantitative and qualitative) suggest authors briefly state some of the salient findings from quantitative studies also.	Already stated in the abstract - Quantitative studies are reported, please see the supplementary files Files 5 & 6, The objective of the NS is to synthesis the findings sections of all included studies that is quantitative, qualitative and mixed method studies.
On page number 2, paragraph 2 line 34 authors talk about life threatening incidents, it may be worth adding what these incidents are on the same page	We give exemplar and refer to the material mortality statistics surveillance reports
references are missing on quite a few places such as line no. 7, 31, and 41	This section is edited
On page number 4 example given in line 20 is not clear and it may help to reword.	We have reworded this
The authors mention that they also categorized foreign students as immigrants which is a different group	This is the definition approved by NIHR in our funding application and the one applied in the review. It is not possible to change this retrospectively as this definition guided our review. It is not a definition that we conceptualised but one that is common usage
In “aim and rationale” section, point “b” is difficult to understand. Can we rephrase it as “women perception about availability of services and their experiences of accessing these services”	Yes we have done this, at your suggestion, however it does mean that this now differs from the final report submitted to NIHR which they have approved

Review Comment	Author Response
In this section authors mentioned that they used Popay's approach to narrative synthesis (NS). It is not clear how this approach is different to traditional NS methods.	Popay's approach has great commonality with other methods of narrative synthesis, however she articulates and specifies the steps with greater coherence. We have mentioned this in the text. Thank for this suggestion.
Authors mentioned that they were interested to include studies testing different interventions; it is not clear why they did not include RCTs.	We did not exclude RCT's and we did not state this anywhere in the manuscript. It is simply no RCT's were identified on the topic with in the review timeframe.
On page 8, authors mention criteria for high, medium and low quality studies but I was not able to find to find how many studies qualified each of these criteria	Yes a very good point and suggestion. Full details are now provided in Table XX
There is no information on how the quantitative data from cross sectional studies and quantitative section of mixed method studies was taken care of	Popay's narrative synthesis approach does not require a focus on date, it requires a focus and synthesis on the narrative findings for each study and we have done this.
On page 10 there is lot of detail under the heading "rigor, reflexivity and quality of synthesis" it will be good if it can be a bit more focused.	We have removed some material from this section.
Reviewer 2	
e.g. lines 7-8 'That is a The UK is in a period of superdiversity defined as a	We have addressed and revised this section, thanks for your suggestions
> "distinguished by a...	

Review Comment	Author Response
The expression and grammar are at times poor; e.g. 'this defers from the former in definition in that the term migrant is conferred on the on the basis of nationality. Meaning all applicants that hold nationality other than the UK are considered migrants. However, the situation is dynamic in that the nationality of a person is may also to change over time and in some individuals may acquire dual-citizenship involving several nation states (lines 17-22).	
We found few interventions that had focused on improving maternity care for these women and the effectiveness of existing interventions have not been rigorously evaluated' this is referring to specific interventions, so you do need to take into account that other interventions which are less specific may also have benefits for this group of women. This is a challenge of doing this kind of review and can be picked up in the article discussion.	Yes we have done this. Please refer to the table of included studies and you will find a number of the studies specified that the study population was a mixed sample not solely focused on immigrant women. However we only included studies with mixed samples where we could clearly identify the findings as they related to immigrant women.
We contacted stakeholders with expertise' referred to. Contacted for what purpose? To identify other studies? To comment on the analysis? ?? and expertise in what?	We have addressed and revised this section, thanks for your suggestion.

Review Comment	Author Response
However, they missed the following published review: > Small R, Roth C, Raval M, Shafiei T, Korfker D, Heaman M, McCourt C, Gagnon A. (2014) Immigrant and non-immigrant women's experiences of maternity care: a systematic and comparative review of studies in five countries. BMC Pregnancy and Childbirth 2014, 14:152 doi:10.1186/1471-2393-14-152	We did not overlook this review we are very familiar with the work of Anita Gagnon and colleagues as the lead author is also a Canadian citizen and worked along side Anita in Canada. The point is a review is not to be include in a systematic review so we presume this comment relates to the background information, so it is not missed in the sense of inclusion in the review. We have mentioned the Small et all study in the background material.
A key issue was that the question stated is quite specifically focused on identifying interventions to improve access; however, this is not really reflected in the study title and abstract and only partly so in the objectives.	Yes this has been a challenge as virtually no interventions are reported in the scientific literature for immigrant women. We discovered via national key-stakeholder event interventions do exist but they have not been rigorously evaluated. At the request of NIHR we amended the review title and we have amended the title for the article. It is difficult to change the aims and objectives as these are what were funded to implement. The fact that they do not exist is a significant review finding.
On pages 8-9 a section of text from 'Data Extraction and assessment of relevance' is repeated word for word in analysis and synthesis.	Addressed
On page 11, lines 3-4 you say, 'Individual team members engaged in independent theming of tabular and coded data.' This is good but can you clarify at what stages and to what extent? All? A sample? How many people?	Yes can provide an exemplar of this - attached

Review Comment	Author Response
The account of studies identified was a little incongruous in that it listed quantitative studies and mixed methods studies, but not the number/type of qualitative studies. For example, contrasting experiences of care are noted with no attempt to identify or explore what kind of factors such as service model or design are associated with more positive or negative experiences of care encounters.	This is an oversight on our part apologies and we have amended this Our included published studies did not necessarily specify the models of service deliver or designs associated with positive and negative care encounters, so we are not able to provide this contextual information.
The statement regarding interpreting and cultural competence of professionals is also not new, so again, the question might be better posed in relation to how this might be improved? the whole article needs considerable editing to improve the grammar, use of punctuation and the expression and proof read to ensure that sentences are actually complete and make sense	Yes we agree and we have included suggestions now how this might be improved We have undertaken a further editorial review of the paper
The way in which the critical appraisal was used was not very clear. Were low quality studies excluded? How did you relate the low/med/high classification to the thick/thin one? At what stage and how did you integrate the CA in the analysis?	We have provided a table which maps our the critical appraisal dimensions, including scientific quality and relevance. See Table

Review Comment	Author Response
Editorial Comments	
he search currently goes to the end of 2017 - are you able to update this?	This is not possible as members of team have left the UoN and no longer have access to the university systems
- Please go through the PRISMA extension for abstracts (http://www.prisma-statement.org/Extensions/Abstracts.aspx) and check that items 1-10 are reported in your abstract.	Retrieved and addressed
- The discussion of strengths and limitations within the Discussion section should be written as a paragraph, rather than inserted as bullet points. We only require the Strengths and Limitations section that is present after the abstract to be formatted into bullet points.	Addressed
- Please include the Patient and Public Involvement statement as part of the Methods section.	Relocated
- Please re-upload your Figure 1 in TIFF, JPG or PDF format and make sure that they have a resolution of at least 300 dpi. Figures in DOCUMENT, EXCEL and POWER POINT format are not acceptable.	Addressed
- The in text citation for "Supplementary file 6" is missing in your main text of your main document file. Please amend accordingly.	Addressed

VERSION 2 – REVIEW

REVIEWER	Reem Salim Malouf University of Oxford
REVIEW RETURNED	01-Aug-2019

GENERAL COMMENTS	This is a good review concerning important and interesting research question Few suggestions: 1) Search strategy : Please note that the search date is more than two years old 2) Results: You included qualitative, quantitative and mixed method studies , but do not state in text which outcomes/.themes that were of interest in these trials. 3) Results: indicate the number and references of studies contributed to each of the five themes. 4) In Fig.2: the starting search date should be 1990 for all databases; 5) General formatting issues and some typos.
--